# ORDER-AWARE INTERACTIVE SEGMENTATION

**Bin Wang**[1,2]\*, **Anwesa Choudhuri**[2], **Meng Zheng**[2], **Zhongpai Gao**[2], **Benjamin Planche**[2],
**Andong Deng**[3], **Qin Liu**[4], **Terrence Chen**[2], **Ulas Bagci**[1], **Ziyan Wu**[2]

[1]Northwestern University, Chicago, IL, USA
[2]United Imaging Intelligence, Boston, MA, USA
[3]University of Central Florida, Orlando, FL, USA
[4]University of North Carolina at Chapel Hill, Chapel Hill, NC, USA
{first.last}@northwestern.edu, {first.last}@uii-ai.com

## ABSTRACT

Interactive segmentation aims to accurately segment target objects with minimal user interactions. However, current methods often fail to accurately separate target objects from the background, due to a limited understanding of *order*, the relative depth between objects in a scene. To address this issue, we propose OIS: order-aware interactive segmentation, where we explicitly encode the relative depth between objects into *order maps*. We introduce a novel order-aware attention, where the order maps seamlessly guide the user interactions (in the form of clicks) to attend to the image features. We further present an object-aware attention module to incorporate a strong object-level understanding to better differentiate objects with similar order. Our approach allows both dense and sparse integration of user clicks, enhancing both accuracy and efficiency as compared to prior works. Experimental results demonstrate that OIS achieves state-of-the-art performance, improving mIoU after one click by 7.61 on the HQSeg44K dataset and 1.32 on the DAVIS dataset as compared to the previous state-of-the-art SegNext, while also doubling inference speed compared to current leading methods.

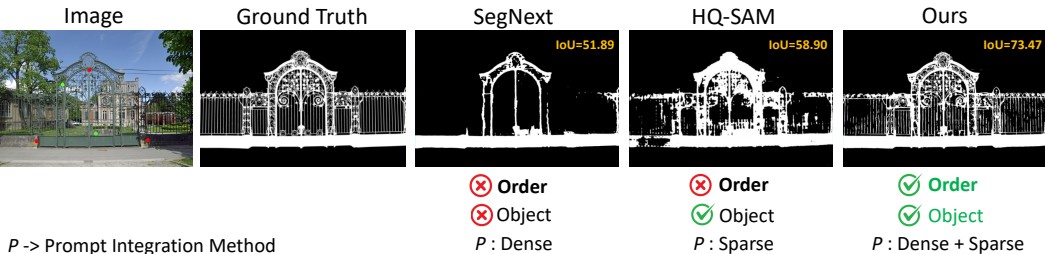

Figure 1: Comparison of our method with current state-of-the-art methods, SegNext (Liu et al., 2024a) and HQ-SAM (Ke et al., 2024), using 5 clicks (red dots represent positive clicks and green dots represent negative clicks). Our method is able to better distinguish the target (the gate) from the background (the trees and the building) and achieve a significantly higher interaction-over-union (IoU). This highlights the effectiveness of our contributions: (a) order (the relative depth between objects), (b) object awareness, and (c) the combination of dense and sparse prompt integration.

## 1 INTRODUCTION

Interactive segmentation allows users to segment objects or regions using prompts, which can be in the form of clicks, scribbles, or bounding boxes. This task is indispensable for applications like image editing (Ling et al., 2021; Abdal et al., 2020), video object tracking (Cheng et al., 2024; Bekuzarov et al., 2023), 3D scene understanding (Yue et al., 2023; Choi et al., 2024), and medical

---

\*This work was done during the internship of Bin Wang at United Imaging Intelligence, Boston, MA.

image annotation (Ma et al., 2024). The goal of this task is to minimize the number of user interactions without compromising on the segmentation accuracy. Achieving this requires the model to have a strong discriminative capability to retain only the foreground features relevant to the object of interest while discarding unrelated background regions, as guided by the user prompts.

Recent methods for interactive segmentation RITM (Sofiiuk et al., 2022), SimpleClick (Liu et al., 2023), SegNext (Liu et al., 2024a), SAM (Kirillov et al., 2023), HQ-SAM (Ke et al., 2024), DynaMITe (Rana et al., 2023)) suffer from three main issues. *Firstly* and most importantly, current methods lack a sense of relative depth between objects in a scene, which makes it difficult for them to distinguish objects accurately. For instance, as illustrated in Figure 1, current state-of-the-art interactive segmentation methods like SegNext (Liu et al., 2024a) and HQ-SAM (Ke et al., 2024) fail to separate the target object (the gate) from the background (trees and buildings). Each object has a specific *order*, or relative depth from each other, corresponding to its location in 3D space (e.g., the gate is in front of the building and trees in Figure 1). Current methods fail to incorporate this information, often leading to inaccurate segmentation masks with many false positives and false negatives (Figure 1). As a result, users are required to perform multiple interactions, i.e., redundant positive clicks to capture the entire foreground and negative clicks to exclude the background. *Secondly*, the encoded positive and negative clicks in current methods interact with the entire image features, causing intermingling of the foreground and background, and preventing a strong discriminative representation of the target object. This is problematic because, intuitively, negative clicks should attend only to the background regions, and positive clicks should only interact with the foreground regions, as investigated in a prior work (Cheng et al., 2024), to allow better discrimination between the target object and background. *Thirdly,* current methods are either computationally inefficient (methods with dense integration of prompts like RITM (Sofiiuk et al., 2022), SimpleClick (Liu et al., 2023), and SegNext (Liu et al., 2024a)) or suffer from misalignment between the image features and prompts (methods with sparse integration of prompts like SAM (Kirillov et al., 2023), HQ-SAM (Ke et al., 2024), and DynaMITe (Rana et al., 2023)).

To address the aforementioned issues, in this work, we propose a novel method called order-aware interactive segmentation (OIS). To address the first issue discussed above, OIS incorporates *order*, i.e., relative depth between objects in a scene. For this, we construct *order maps* by utilizing an additional depth map and calculating the relative depth between each point in the image and the user prompt's location. As illustrated in Figure 3, the order map effectively captures the relative depths between the target object, indicated by the user's prompt, and other regions in the image. To efficiently integrate order maps in our framework, we propose a novel *order-aware attention*, which guides the model to focus more on regions close to the prompt's order and less on regions that are farther from the prompt order. This enables the model to better understand the target object's position in 3D space, improving its ability to distinguish the target from surrounding objects.

While order-aware attention helps differentiate target objects at different depths, it struggles to separate objects with similar depths. To address this and the second issue discussed above, we incorporate the notion of objects through *object-aware attention*, a novel prompt-based masked cross-attention mechanism. It ensures that positive and negative prompts are separated to align with their own corresponding regions. The foreground and background prompts are encoded into respective foreground and background embeddings that attend to different foreground and background regions. Note that explicit foreground-background attention has been used previously for the video object segmentation task (Cheng et al., 2024), but to the best of our knowledge, we are the first to use this explicit separation for interactive segmentation. Unlike Cheng et al. (2024), we encode the foreground and background clicks into foreground and background embeddings to introduce an explicit notion of the target object and enable the network to distinguish different objects with similar depth.

To address the third issue, we combine both dense and sparse integration of the prompts, improving accuracy and efficiency. We retain the superior spatial alignment of dense integration methods, while being as computationally efficient as sparse integration methods.

In summary, the contributions of our proposed OIS are listed as follows:

- We propose order-aware attention, which integrates order, or relative depths between objects, into interactive segmentation, improving the model's ability to distinguish target objects based on their relative depths from one another.

- We introduce object-aware attention to incorporate a strong understanding of objects. This enables our model to better differentiate objects with similar order.

- Our approach combines both dense and sparse integration of prompts. This novel design improves the alignment between the image and prompts while maintaining computational efficiency.

- Experimental results show that our OIS delivers state-of-the-art performance with low latency, improving 7.61 mIoU after one click one click on HQSeg44K (Ke et al., 2024), 1.32 on DAVIS (Perazzi et al., 2016), and 2 times faster than the current best method, SegNext (Liu et al., 2024a).

## 2 RELATED WORK

**Interactive segmentation.**  Current methods for interactive segmentation can be categorized into two groups: dense fusion and sparse fusion. Dense fusion includes RITM (Sofiiuk et al., 2022), FocalClick (Chen et al., 2022), SimpleClick (Liu et al., 2023), InterFormer (Huang et al., 2023), and SegNext (Liu et al., 2024a). While sparse fusion mainly involves DynaMITe (Rana et al., 2023), SAM (Kirillov et al., 2023), HR-SAM (Huang et al., 2024), and HQ-SAM (Ke et al., 2024). In dense fusion, user prompts are represented as dense maps and aligned with the image using attention mechanisms. However, this approach increases inference time significantly due to the costly attention between spatial feature maps (image and prompt dense map features). In contrast, sparse fusion converts user prompts into sparse embeddings, which lowers computational costs during cross-attention with image features. However, this efficiency comes at the cost of reduced alignment precision between the prompts and the image. We combine the best of both types of fusion, which retains the spatial alignment capabilities of dense fusion while improving the efficiency through sparse fusion. More importantly, all these methods lack relative 3D information, which makes it difficult to accurately distinguish objects in complex scenarios. To address this issue, we introduce order information and order-aware attention to incorporate 3D spatial context, enhancing the prompt's ability to separate targets more effectively. More related works can be found in Appendix A.3.

**Depth in segmentation tasks.**  Many image and video segmentation tasks have utilized depth as an additional modality to better handle foreground-background separation, such as instance segmentation (Wu et al., 2023), semantic segmentation (Zhang et al., 2023b; Yin et al., 2023), panoptic segmentation (Gao et al., 2022), video object segmentation (Liu et al., 2024b), video panoptic segmentation (Yuan et al., 2022). However, depth has hardly been explored for interactive segmentation. The only related work, MM-SAM (Wang et al., 2022), adopts an extra encoder to integrate depth. However, they achieve poor performance in common interactive segmentation benchmarks due to the insufficient fusion between prompt and depth. In this work, instead of directly inputting depth, we introduce *order map* that combines both depth and user prompt to further strengthen the model sense of relative depth between objects.

**Object understanding in segmentation tasks.**  Recent transformer-based approaches like DETR (Carion et al., 2020), MaskFormer (Cheng et al., 2021), Mask2Former (Cheng et al., 2022), ViSTR (Wang et al., 2021), HODOR (Athar et al., 2022), etc, introduce the concept of abstract object queries to represent objects in images and videos and seamlessly perform tasks like object detection and segmentation. Cutie (Cheng et al., 2024) further enhances the object representations for the video object segmentation task by having separate foreground and background regions that the object queries attend to, allowing the queries to distinguish between the foreground and background. However, the aforementioned methods were not directly applicable to the interactive segmentation task. To address this issue, SAM (Kirillov et al., 2023) introduces a sense of objects into the interactive segmentation task by encoding positive and negative prompts into sparse embeddings, all of which together represent the target object. However, SAM fails to clearly distinguish between the foreground and background because both the positive and the negative embeddings attend to the same regions, unlike Cutie. To address this limitation of SAM, we introduce a foreground-background separation into interactive segmentation, referred to as object-aware attention. It enables the sparse embeddings to only attend to their corresponding regions based on their type (positive click embeddings attend to the foreground and negative click embeddings attend to the background).

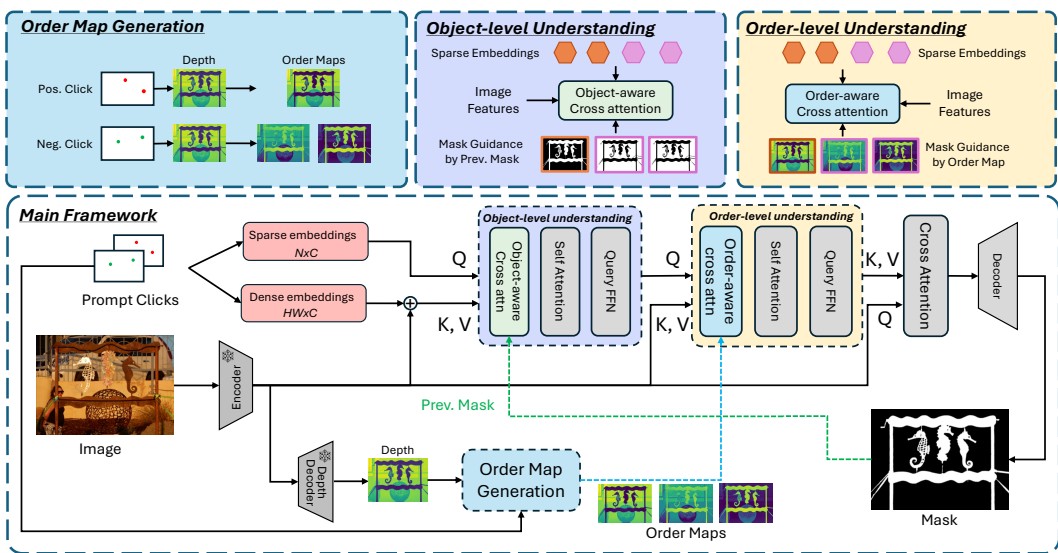

Figure 2: Overview of our proposed OIS framework. Our order maps are generated (blue box on the top-left) to capture the relative depth of objects in a scene, as described in Sec. 3.2. The order maps selectively guide the sparse embeddings to attend to the image features in our novel order-aware attention module (highlighted in blue inside the order-level understanding block), as described in Sec. 3.2. The object-aware attention module (highlighted in green in the object-level understanding block) imparts a strong discriminative notion of objects, as discussed in Sec. 3.3. We utilize both sparse and dense integration of prompts (highlighted in red), as described in Sec. 3.4.

# 3 METHODOLOGY

## 3.1 OVERVIEW

The framework of our proposed OIS is depicted in Figure 2. Given an image with a target object, the image features are first extracted using an image encoder. Following prior works (Huang et al., 2023; Kirillov et al., 2023), the image is encoded only once to save computational cost. The same extracted image features are reused for subsequent interactive stages when new user clicks are added. Each new user click is encoded in two ways, including dense and sparse embeddings (Sec. 3.4). Dense embeddings are added to the image features to enhance prompt alignment, while sparse embeddings interact with the image features through two types of attention mechanisms: a) our proposed order-aware attention (Sec. 3.2) and b) object-aware attention (Sec. 3.3). This enables the model to contain a sense of relative depth and the notion of objects. Finally, the enhanced features are passed through a decoder to predict the target segmentation mask.

## 3.2 ORDER-LEVEL UNDERSTANDING

**Order map.** To make the model have a sense of relative depth, we first introduce the concept of order, which represents the relative depth between objects in a scene. We formulate this information as an order map, defined as the relative distance in camera optical axis direction. It is generated by combining the depth map $\mathcal{R} \in \mathbb{R}^{H \times W}$ of the image with each interactive prompt. Let $\Omega^p$ indicate the set of positive prompts, and $\Omega^n$ be the set of negative prompts. For the $i$th positive prompt $p_i^p \in \Omega^p$, the order map $\mathcal{M}^p$ is defined as:

$$\mathcal{M}^p = |\mathcal{R} - \frac{1}{M} \sum_{p_i^p \in \Omega^p} \mathcal{R}_{p_i^p}|, \tag{1}$$

where $\mathcal{R}_{p_i^p}$ is the depth value at the coordinate of the positive prompt, and $M$ is the number of positive prompts. Notice that since all the positive prompts refer to the same object, they share a common order map, which means order map $\mathcal{M}^p$ is the same for all positive prompts $p_i^p \in \Omega^p$.

Figure 3: Illustration of order map (after normalization into 0-1). Red dots in (c) indicate positive prompt clicks, and white dots in (d) and (e) represent negative prompt clicks. In order maps, darker means closer to prompt-selected object, while lighter areas are farther to the prompt-selected object.

However, in contrast, each negative prompt can correspond to different objects in the background, so each of them has its own order map. For the $i$th negative prompt $p_i^n \in \Omega^n$, the corresponding order map $\mathcal{M}_i^n$ is expressed as:

$$\mathcal{M}_i^n = |\mathcal{R} - \mathcal{R}_{p_i^n}|, \tag{2}$$

where $\mathcal{R}_{p_i^n}$ is the depth value at the coordinate of the negative prompt. In Figure 3, it is difficult to distinguish the foreground sculpture frame from the background ball using the image alone. But with the help of order map, it is easier to use prompt to separate them. In Figure 3(c), the foreground frame is fully separated from the background, and in Figure 3(d), the background ball is also clearly distinguished with a negative prompt. This means that if false positives occur on the background ball, one negative prompt can now recognize the entire ball, reducing unnecessary interactions to remove false positives.

**Order-aware attention.** The order map, proposed above, guides the sparse embeddings to attend to the image features via our novel *order-aware attention*. In sparse embeddings $\mathcal{S} \in \mathbb{R}^{N \times C}$, the first half of $N$ embeddings are generated from the positive prompts and the second half from the negative prompts. We construct order mask $\mathcal{M} \in (0, 1)^{N \times HW}$ as $[\mathcal{M}^p, \mathcal{M}^p, ..., \mathcal{M}_1^n, \mathcal{M}_2^n ...]$, where the first half contains order maps shared by all positive prompts, and the second half consists of individual order maps for each negative prompt. In order-aware attention, the goal is to make the prompt aware of the order within the image. To achieve this, we update the sparse embeddings by performing masked cross-attention with the image features, where the order mask guides the attention to relevant regions. The order-aware attention is defined as:

$$\mathcal{S}' = \text{softmax}(QK^T - \sigma\mathcal{M})V + \mathcal{S}, \tag{3}$$

where $Q$ is the linear transformation of sparse embedding $\mathcal{S}$, and $K, V$ are the linear transformation of image features $\mathcal{F}$. Inspired by Athar et al. (2022), $\sigma\mathcal{M} \in (0, +\infty)$ is added to control the attention with a learnable scale parameter $\sigma$. Specifically, if a region is close to the prompt-selected order (i.e., its corresponding order mask value is near 0), it behaves like normal cross attention. However, if the region is far from the prompt-selected order (i.e., its order mask value approaches $+\infty$ after scaling), the attention weight approaches 0 due to softmax, indicating minimal attention. In summary, regions close to the prompt-selected order receive higher attention, while far regions receive less.

### 3.3 OBJECT-LEVEL UNDERSTANDING

Although order-level understanding can guide the prompt in distinguishing objects from different orders, it faces the limitation of differentiating between objects within the same order. To solve this problem, we introduce object-aware attention to impart a sense of objects into our model.

**Object-aware attention.** After going through order-aware attention, the sparse embeddings (Sec. 3.2) attend to the image features using a foreground-background separated masked cross-attention following Cutie (Cheng et al., 2024). We call this our *object-aware attention* module. Specifically, in this module, the sparse embeddings generated from the positive clicks attend only to the foreground regions, while the sparse embeddings generated from the negative clicks focus only on the background regions. The foreground and background regions are obtained from the prediction generated using the previous clicks. This explicit foreground-background separation introduces a strong notion of the target object into the sparse embeddings. Different from Cutie (Cheng et al., 2024), where multiple object queries are randomly initialized, we encode the foreground and background clicks to generate sparse embeddings, which act as attributes of one single target object.

| Methods | Backbone | NoC90 ↓ | NoC95 ↓ | 1-mIoU ↑ | 5-mIoU ↑ | NoF95 ↓ |
|---|---|---|---|---|---|---|
| RITM (Sofiiuk et al., 2022) | HRNet32$_{400}$ | 10.01 | 14.58 | 36.03 | 77.72 | 910 |
| FocalClick (Chen et al., 2022) | SegF-B3-S2$_{256}$ | 8.12 | 12.63 | 62.89 | 84.63 | 835 |
| FocalClick (Chen et al., 2022) | SegF-B3-S2$_{384}$ | 7.03 | 10.74 | 61.92 | 85.45 | 649 |
| SimpleClick (Liu et al., 2023) | ViT-B$_{448}$ | 7.47 | 12.39 | 65.54 | 85.11 | 797 |
| InterFormer (Huang et al., 2023) | ViT-B$_{1024}$ | 7.17 | 10.77 | 64.40 | 82.62 | 658 |
| SAM (Kirillov et al., 2023) | ViT-B$_{1024}$ | 7.46 | 12.42 | 45.08 | 86.16 | 811 |
| EiffcientSAM (Xiong et al., 2024) | ViT-T$_{1024}$ | 10.11 | 14.60 | - | 77.90 | - |
| EiffcientSAM (Xiong et al., 2024) | ViT-S$_{1024}$ | 8.84 | 13.18 | - | 79.01 | - |
| MobileSAM (Zhang et al., 2023a) | ViT-T$_{1024}$ | 8.70 | 13.83 | 53.20 | 81.98 | 951 |
| HQ-SAM (Ke et al., 2024) | ViT-B$_{1024}$ | 6.49 | 10.79 | 42.38 | 89.85 | 671 |
| HR-SAM (Huang et al., 2024) | ViT-B$_{1024}$ | 5.42 | 9.27 | - | 91.81 | - |
| HR-SAM++ (Huang et al., 2024) | ViT-B$_{1024}$ | 5.32 | 9.18 | - | 91.84 | - |
| SegNext (Liu et al., 2024a) | ViT-B$_{1024}$ | 5.32 | 9.42 | 81.79 | 91.75 | 583 |
| Ours | ViT-B$_{1024}$ | **3.95** | **7.50** | **89.40** | **93.78** | **485** |
| EiffcientSAM (Xiong et al., 2024) | ViT-T$_{2048}$ | 9.47 | 13.13 | - | 74.20 | - |
| EiffcientSAM (Xiong et al., 2024) | ViT-S$_{2048}$ | 8.27 | 11.97 | - | 74.91 | - |
| HR-SAM (Huang et al., 2024) | ViT-B$_{2048}$ | 4.37 | 7.86 | - | 93.34 | - |
| HR-SAM++ (Huang et al., 2024) | ViT-B$_{2048}$ | 4.20 | 7.79 | - | 93.32 | - |
| Ours | ViT-B$_{2048}$ | **3.47** | **6.63** | **89.57** | **94.45** | **398** |

Table 1: Performance comparison on the HQSeg44K benchmark.

Formally, the object-aware attention is defined as:

$$\mathcal{S}' = \text{softmax}(QK^T + \mathcal{H})V + \mathcal{S}, \tag{4}$$

where $Q$ is the linear transformation of sparse embedding $\mathcal{S}$, and $K, V$ are the linear transformation of image features $\mathcal{F}$. Here, Mask $\mathcal{H}$ is

$$\mathcal{H}(x,y) = \begin{cases} 0 & \text{if } H(x,y) = 1 \\ -\infty & \text{otherwise} \end{cases}. \tag{5}$$

We construct mask $H \in \{0,1\}^{N \times HW}$ as $[H^f, H^f, ..., \hat{H}^f, \hat{H}^f, ...]$, where the first half consists of the previous mask $H^f$ (with 1 indicating the foreground target and 0 for the background), and the second half consists of the inverse previous mask $\hat{H}^f$ (with 1 indicating the background and 0 for the foreground target). Note that in the first round of interaction, when no previous mask is available, we replace object-aware attention with the standard cross-attention proposed in (Vaswani, 2017).

## 3.4 PROMPT INTEGRATION VIA DENSE AND SPARSE FUSION

Unlike previous methods that encode prompts through either dense or sparse fusion, we leverage the strengths of both approaches while addressing their limitations. Dense fusion can preserve more spatial details from the prompt (Liu et al., 2024a). By adding the dense embeddings $\mathcal{D} \in \mathbb{R}^{H \times W \times C}$ to the image features $\mathcal{F} \in \mathbb{R}^{H \times W \times C}$, the prompts achieve better spatial alignment with the image. Typically, this process is followed by a self-attention module (Liu et al., 2024a) to capture spatial relationships. However, direct attention between spatial features is computationally expensive. To address this issue, we remove the self-attention module and introduce the sparse embeddings $\mathcal{S} \in \mathbb{R}^{N \times C}$, where $N$ is the number of prompts. Sparse fusion is more computationally efficient (Cheng et al., 2024), as the attention is applied between sparse embeddings and spatial features. Moreover, sparse fusion offers greater flexibility for performing attention operations. Leveraging this flexibility, we have developed order-aware and object-aware attention mechanisms that enable the model to focus on specific regions based on the sequence of user interactions or the contextual understanding of objects within the image. Thus, our approach improves the alignment between the image and prompt through dense embeddings while maintaining computational efficiency through sparse fusion and simultaneously leverages its flexibility to make the prompt aware of order and object understanding.

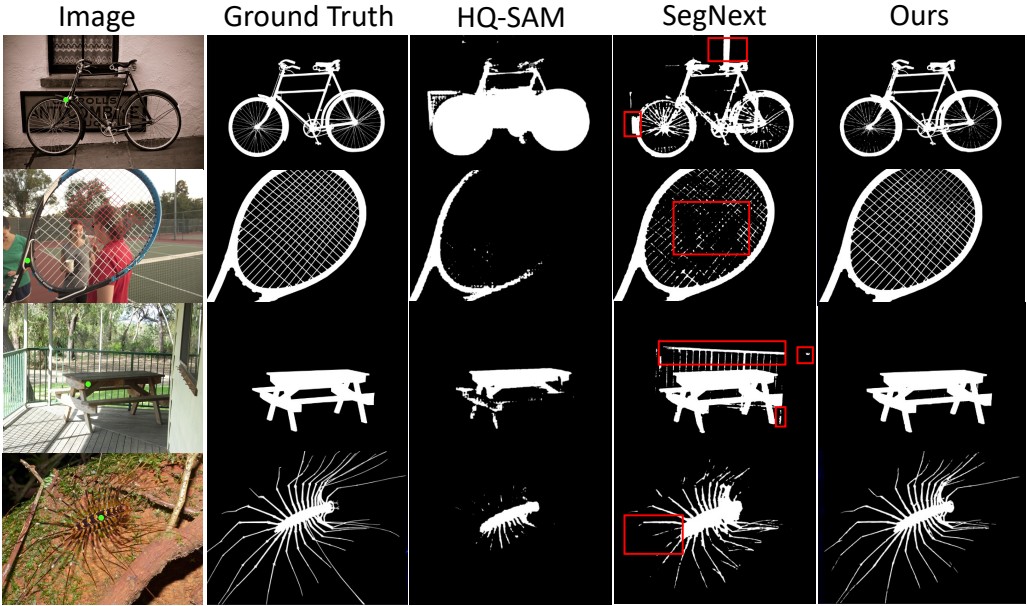

Figure 4: Qualitative results on HQSeg44K. Green dots indicate the user's first clicks.

## 4 EXPERIMENTS

### 4.1 EXPERIMENT SETTINGS

**Datasets and evaluation metrics.** We evaluate our method and comparison methods on two widely used benchmarks for interactive segmentation: HQSeg44K (Ke et al., 2024) and DAVIS (Perazzi et al., 2016). More details about these two datasets can be found in Appendix A.2. We report five evaluation metrics: number of clicks (NoC), mean interaction-over-union (mIoU), number of failures (NoF), seconds per click (SPC), and SAT latency. The NoC metrics, including NoC90 and NoC95, represent the number of clicks required to reach mIoU thresholds of 90% and 95%, respectively. The mIoU metrics, such as 1-mIoU and 5-mIoU, denote the average IoU achieved after 1 or 5 consecutive clicks. NoF measures the number of cases that require more than 20 clicks to reach 90% mIoU. SPC evaluates time efficiency by calculating the average time taken per click, while SAT latency refers to the total latency for the Segment Anything Task (with a grid of 16×16 points). For comparison methods, we utilize their released model weights to compute the 1-mIoU and NoF95 scores. For all other metrics, we directly reference the results as reported in their papers.

**Implementation details.** In our framework, we use a frozen ViT-Base encoder from DepthAnythingV2 (Ranftl et al., 2021) to encode each image. Note that we re-use the same encoder for depth map generation, allowing us to keep our overall model lightweight (as shown in Table 3). The encoder is kept frozen during training to retain the depth features. We have 2 decoders initialized from DepthAnythingV2 (Ranftl et al., 2021): one is kept frozen to generate the depth map and the other one is trained to generate the final segmentation. We formulate the dense embeddings (introduced in Sec. 3.4) by constructing a two-channel dense map (one for positive clicks and the other for negative clicks). Then, we encode the dense map via a single convolution layer. For sparse embeddings, we represent each click as a $C$-dimensional embedding ($C = 128$) following SAM (Kirillov et al., 2023). Each of these embeddings is the sum of the positional encoding of its coordinate and a learnable embedding based on its prompt type (positive or negative). We set the maximum number of clicks $N$ as 48 during training, with the first 24 for positive clicks and the remaining for negative. If there are fewer than 48 clicks, we fill the remaining slots with an additional 'non-point' embedding. We use three same blocks of object-level and order-level understanding modules (Sec. 3.2 and Sec. 3.3) to conduct prompt fusion. In these modules, the Feed-Forward Network (FFN) is implemented as a 2-layer Multi-Layer Perceptron (MLP). Each attention module is followed by a LayerNorm, which normalizes the sparse embeddings.

| Methods | Backbone | NoC90 ↓ | NoC95 ↓ | 1-mIoU ↑ | 5-mIoU ↑ | NoF95 ↓ |
|---|---|---|---|---|---|---|
| RITM (Sofiiuk et al., 2022) | HRNet32$_{400}$ | 5.34 | 11.45 | 72.53 | 89.75 | 139 |
| FocalClick (Chen et al., 2022) | SegF-B3-S2$_{256}$ | 5.17 | 11.42 | 76.28 | 90.82 | 155 |
| FocalClick (Chen et al., 2022) | SegF-B3-S2$_{384}$ | 4.90 | 10.40 | 76.35 | 91.22 | 123 |
| SimpleClick (Liu et al., 2023) | ViT-B$_{448}$ | 5.06 | 10.37 | 72.90 | 90.73 | 107 |
| InterFormer (Huang et al., 2023) | ViT-B$_{1024}$ | 5.45 | 11.88 | 64.40 | 87.79 | 150 |
| SAM (Kirillov et al., 2023) | ViT-B$_{1024}$ | 5.14 | 10.74 | 48.66 | 90.95 | 154 |
| MobileSAM (Zhang et al., 2023a) | ViT-T$_{1024}$ | 5.83 | 12.74 | 61.69 | 89.18 | 196 |
| EifficientSAM (Xiong et al., 2024) | ViT-T$_{1024}$ | 7.37 | 14.28 | - | 85.26 | - |
| EifficientSAM (Xiong et al., 2024) | ViT-S$_{1024}$ | 6.37 | 12.26 | - | 87.55 | - |
| HQ-SAM (Ke et al., 2024) | ViT-B$_{1024}$ | 5.26 | 10.00 | 45.75 | 91.77 | 136 |
| HR-SAM (Huang et al., 2024) | ViT-B$_{1024}$ | 4.82 | 11.86 | - | 91.34 | - |
| HR-SAM++ (Huang et al., 2024) | ViT-B$_{1024}$ | 5.02 | 11.64 | - | 91.25 | - |
| SegNext (Liu et al., 2024a) | ViT-B$_{1024}$ | 4.43 | 10.73 | 85.97 | 91.87 | 123 |
| Ours | ViT-B$_{1024}$ | **3.80** | **8.59** | **87.29** | **92.76** | 114 |
| EifficientSAM (Xiong et al., 2024) | ViT-T$_{2048}$ | 8.00 | 14.37 | - | 84.10 | - |
| EifficientSAM (Xiong et al., 2024) | ViT-S$_{2048}$ | 6.86 | 12.49 | - | 85.17 | - |
| HR-SAM (Huang et al., 2024) | ViT-B$_{2048}$ | 4.22 | 8.83 | - | 92.63 | - |
| HR-SAM++ (Huang et al., 2024) | ViT-B$_{2048}$ | 4.12 | 8.72 | - | 92.73 | - |
| Ours | ViT-B$_{2048}$ | **3.48** | **8.42** | **88.05** | **92.90** | 105 |

Table 2: Performance comparison on the DAVIS benchmark.

**Training and evaluation.** To have a fair comparison, we follow the prior works (Sofiiuk et al., 2022; Liu et al., 2023; Huang et al., 2023; Liu et al., 2024a) by applying the same click sampling strategies for samples in training and evaluation, and adopting normalized focal loss as our loss function. Details of the click strategy are in Appendix A.1. The input image size is set to 1024. We use Adam optimizer to train our model on HQSeg44K dataset for 15 epochs on two A100 GPUs.

## 4.2 EVALUATION ON HQSEG44K

We first evaluate on HQSeg44K dataset (Ke et al., 2024). As shown in Table 1, our method significantly outperforms other methods across all the metrics. In particular, our NoC90 score shows an improvement of more than 1 click, while our NoC95 score improves by approximately 2 clicks. Furthermore, our 1-mIoU achieves a remarkable improvement of around 8 points compared to SegNext. This indicates that with only one click, our model can already provide an accurate and good-quality segmentation mask with mIoU of 89.40% compared to 81.79% of SegNext.

In Figure 4, we present some qualitative results from the HQSeg44K dataset with one user click input. In the first example, we observe that SegNext is negatively impacted by the background because the bike and window have similar color, leading to parts of the background being mistakenly classified as the target object. In contrast, our model accurately segments the bike without any false positives from the background. The second example highlights this issue even more clearly. SegNext struggles to segment the racket strings, as it is confused by the humans behind the racket. However, our method effectively alleviates this problem, successfully segmenting the strings without interference. In the third case, SegNext includes false positives from both background railing and the wall in front of the table, while our model avoids these errors entirely. This suggests that our method has a superior ability to distinguish the spatial position of target objects in 3D space. Overall, these examples demonstrate that the introduction of order information effectively reduces false positives. (We also provide multi-round click results in Appendix A.7 and challenging cases in Appendix A.6.)

## 4.3 EVALUATION ON DAVIS

Table 2 provides the comparison of different methods on the DAVIS (Perazzi et al., 2016) dataset. Our method achieves the state-of-the-art performance in both NoC and mIoU metrics. Specifically, our model requires only 8.59 clicks to reach 95% mIoU, compared to 10.73 clicks for SegNext, which is over 2 clicks improvement. This means that even in complex scenarios, our method can predict high-quality segmentation masks and generalize well. Figure 5 further illustrates this point.

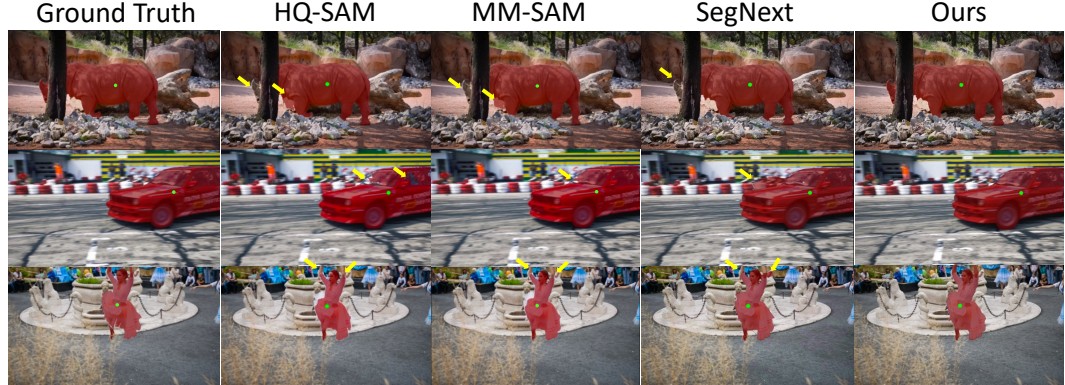

Figure 5: Qualitative result on DAVIS. Green dots indicate the user's first clicks.

As shown in Figure 5, SegNext is notably affected by background objects, image blur, occlusion, and complex background. In the first case, the rhinoceros is occluded by the tree and separated into two parts. SegNext is only able to segment the part of the rhinoceros where a click is placed. Our approach, however, successfully segments the entire rhinoceros with just 1 click, highlighting our order- and object-level understanding. In the second case, due to the fast movement of the car, the image is blurred, especially in the background and around the vehicle. SegNext incorrectly segments the regions surrounding the car as part of the prediction. In contrast, our method handles the image blur effectively. In the third case, due to numerous background objects around the human arm, SegNext fails to fully segment the arm, and also mistakenly includes parts of the background as false positives. We also display the results for SAM-based methods, including HQ-SAM (Ke et al., 2024) and MM-SAM (Xiao et al., 2024). Notice that MM-SAM incorporates depth information via an additional depth encoder. However, despite this enhancement, it still performs similarly to SegNext and fails in these cases. Additional depth integration comparisons are in Appendix A.4.

## 4.4 EFFICIENCY ANALYSIS

| Methods | Parameters (M) | SPC (ms) ↓ | SAT Latency (s) ↓ | NoC90 ↓ | 5-mIoU ↑ |
|---|---|---|---|---|---|
| SimpleClick | 96.46 | 55 | 81.3 | 7.47 | 85.11 |
| HQ-SAM | 94.81 | **10** | **5.1** | 6.49 | 89.85 |
| SegNext | 113.79 | 58 | 20.6 | 5.32 | 91.75 |
| Ours | 107.88 | 31 | 9.2 | **3.95** | **93.78** |

Table 3: Efficiency and accuracy comparison of interactive segmentation methods. Our model achieves the best balance, offering low latency with superior segmentation accuracy.

Time efficiency is critical in interactive segmentation models, as users expect minimal latency to ensure smooth and responsive interactions. In Table 3, we compare our methods with leading approaches in terms of efficiency (SPC and SAT Latency) and accuracy (NoC90 and 5-mIoU). The SPC (ms) reflects the average speed per click, while SAT Latency (s) captures the overall time, accounting for image processing, depth model prediction, click processing, and segmentation.

SimpleClick is the slowest in terms of SPC, primarily because it re-encodes the image after each new user click. This introduces significant time costs that increase its overall SAT latency. In contrast, our method encodes the image only once, regardless of the number of subsequent clicks, similar to HQ-SAM and SegNext. As a result, we exhibit lower SPC and SAT Latency.

Among these, although HQ-SAM is slightly faster, its accuracy is significantly lower than our model. SegNext, while being slightly less accurate than ours, performs 2 times slower than our method in terms of SPC and SAT latency. This discrepancy arises because our method computes cross attention between sparse queries and spatial image feature, replacing the heavy self-attention between high resolution spatial features used in SegNext. In summary, our method achieves the optimal balance between efficiency and accuracy, offering low latency while achieving superior segmentation quality.

Figure 6: Visualization of attention weights before and after applying order-aware attention. Attention weights initially spread across overlapping objects with the same semantics, but after applying order-aware attention, they focus solely on the target object.

## 4.5 ABLATIONS

To better understand how each proposed module performs, we conduct the ablation experiments here. Table 4 demonstrates the importance of each module in the proposed method by analyzing performance across three different kinds of metrics (NoC90, 5-mIoU, and NoF95) on DAVIS dataset.

**Importance of order-aware attention.** The 2nd row in Table 4 shows the effect of removing the order-aware attention module. We observe a significant increase in NoC90 by 1.04 and a decrease in 5-mIoU by 1.15. This indicates the model requires more user interaction without order information. This validates our claim that order-aware attention helps distinguish between objects at different depth levels. This is also highlighted in Figure 6 with two examples where similar objects are overlapping. We visualize the attention weights before and after integrating order-aware attention. We observe that before integrating order-aware attention, the attention weights are distributed across both objects. After incorporating the order information, the attention weights focus solely on the target object. More visualizations of attention weight are in Appendix A.5.

| Methods | NoC90 ↓ | 5-mIoU ↑ | NoF95 ↓ |
|---|---|---|---|
| Full | **3.80** | **92.76** | **114** |
| w/o order | 4.84 (+1.04) | 91.61 (-1.15) | 171 |
| w/o object | 3.92 (+0.12) | 92.52 (-0.24) | 125 |
| w/o sparse | 5.18 (+1.38) | 89.81 (-2.95) | 167 |
| w/o dense | 4.63 (+0.83) | 91.31 (-1.45) | 188 |

Table 4: Ablation experiments on DAVIS.

**Importance of object-aware attention.** The 3rd row in Table 4 shows the effect of removing the object-aware attention. The result shows a performance decrease, indicating that object-aware attention is indispensable.

**Importance of sparse embeddings.** The 4th row of Table 4 shows the performance after completely removing the sparse embedding discussed in Sec. 3.4. This configuration automatically removes both the order-aware and object-aware attention. The prompt integration is solely via the addition of dense embeddings to the image embeddings (Sec. 3.4). We observe a significant performance drop, demonstrating the critical role of the sparse fusion module in effective prompt integration.

**Importance of dense embeddings.** The 5th row of Table 4 shows the effect of removing the dense embeddings (Sec. 3.4). The prompt is only represented by the sparse embeddings in this configuration. This results in a large performance decline, as dense fusion improves spatial alignment between the image and prompt, making prompt interactions more effective.

## 5 CONCLUSION

In this work, we have introduced Order-Aware Interactive Segmentation (OIS), which incorporates 3D spatial context through the concept of order into 2D interactive segmentation. By leveraging proposed order-aware attention, the model can better distinguish objects based on their relative depths. We introduced object-aware attention to enhance our model's ability to differentiate objects within the same depth level. We integrated user clicks using both sparse and dense representations, improving segmentation accuracy and computational efficiency. Experimental results validated that OIS significantly improves segmentation accuracy and speed as compared to prior methods.

**Ethics Statement.** We have utilized publicly available datasets for training and evaluation, ensuring compliance with all relevant data usage policies and privacy regulations. All datasets used are anonymized to protect individual privacy, and no personally identifiable information is included in our model or results. The model is designed for general-purpose image segmentation tasks; however, we recognize that its deployment in specific applications may raise ethical considerations. We advocate for the responsible use of our technology to prevent misuse in contexts that could infringe on privacy or other ethical standards. Besides, all methodologies and experiments have been conducted with integrity, adhering to established research ethics guidelines. We are committed to transparency and reproducibility, providing thorough documentation of our processes to facilitate independent verification and ethical scrutiny. Finally, we recognize the importance of ongoing dialogue regarding the ethical implications of advancements in image segmentation technologies and encourage the research community to engage in continuous evaluation to uphold ethical standards in future developments.

**Reproducibility Statement.** To ensure the reproducibility of our work, we extensively describe the implementation details, such as the pre-trained model, training data, and other important training hyperparameters in Sec 3, Sec 4.1 and Sec A.1. We will release all source code pending release approval.

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

# A APPENDIX

## A.1 CLICK SAMPLING STRATEGY.

To have fair comparison, we apply the same click sampling strategy from prior works (Sofiiuk et al., 2022; Liu et al., 2023; Huang et al., 2023; Liu et al., 2024a) for evaluation in all the experiments. This strategy generates clicks sequentially, with each new click placed at the center of the largest error region in the model's prediction.

## A.2 MORE DATASETS DETAILS

We evaluate our method and comparison methods on two widely used benchmarks for interactive segmentation: HQSeg44K (Ke et al., 2024) and DAVIS (Perazzi et al., 2016). HQSeg44K is a large-scale segmentation dataset containing 44320 images with high-quality mask labels. It contains a diverse range of images spanning over 1,000 semantic classes, covering both simple and complex scenarios, and includes objects with thin shapes as well as more straightforward forms. DAVIS is a high-precision video object segmentation dataset consisting of 50 videos. It contains more complicated scenarios such as occlusion, multi-objects, motion blurs and many other challenges. To be consistent with previous works, we use a subset of 345 frames to conduct the evaluation.

## A.3 ADDITIONAL RELATED WORK ON INTERACTIVE SEGMENTATION

Early interactive segmentation methods (Boykov & Jolly, 2001; Grady, 2006; Rother et al., 2004; Gulshan et al., 2010) relied on optimization techniques to solve cost functions defined by image pixels. As deep learning became more popular, approaches began incorporating user interactions directly into neural networks (Xu et al., 2016; Sofiiuk et al., 2020; Maninis et al., 2018; Jang & Kim, 2019; Lin et al., 2020). Further advancements such as RITM (Sofiiuk et al., 2022; Liu et al., 2022) leveraged large-scale data for robustness, FocalClick (Chen et al., 2022) introduced local refinement modules, and SimpleClick (Liu et al., 2023) improved performance using vision transformers. OACE (Mathur et al., 2024) is proposed to improve the foreground distinction in a contrastive learning manner. To enhance efficiency, methods like InterFormer (Huang et al., 2023) reduced interaction time by encoding the image only once during interactions, and SegNext (Liu et al., 2024a) applied similar idea with attention mechanism to achieve high quality with low latency. Following the introduction of SAM (Kirillov et al., 2023), numerous SAM-based methods have been proposed to improve interactive segmentation in aspects such as quality (Ke et al., 2024), interaction efficiency (Zhang et al., 2023a), and high-resolution image handling (Huang et al., 2024). Variants of interactive segmentation, like multi-object segmentation (Yue et al., 2023; Rana et al., 2023), medical image segmentation (Ma et al., 2024; Wong et al., 2023), and segmentation with controllable granularity (Zhao et al., 2024), have also been proposed.

## A.4 COMPARISON WITH OTHER INTERACTIVE SEGMENTATION METHODS INCORPORATING 3D INFORMATION

In this section, we compare our methods to other interactive segmentation methods that integrate 3D information (MM-SAM (Xiao et al., 2024)) to show that naively incorporating depth is not as effective as our order-aware attention module discussed in Sec. 3.2 and validated in Sec. 4. Here, we take the depth maps generated from DepthAnything V2 (Yang et al., 2024a) as an input to MM-SAM and test this on the DAVIS dataset; we call this configuration MM-SAM in Table 5. Moreover, following the strategy of MM-SAM, which adds an extra encoder for other modalities such as lidar, depth, and thermal, we also train an additional Depth Encoder and add to our pipeline. Note that in this configuration, we disable the order-aware attention discussed in 3.2 to ensure fair comparison. We call this configuration DepthEncoder in Table 5. Our method outperforms this configuration and MM-SAM across all metrics. This indicates that incorporating 3D information into the model through order using our specialized order-aware attention module is a better solution than directly integrating depth maps.

| Methods | NoC90 ↓ | NoC95 ↓ | 1-mIoU ↑ | 5-mIoU ↑ | NoF95 ↓ |
|---|---|---|---|---|---|
| MM-SAM (Xiao et al., 2024) | 6.22 | 12.26 | 51.41 | 88.19 | 181 |
| DepthEncoder | 3.88 | 10.19 | 85.36 | 92.15 | 140 |
| Ours | **3.80** | **8.59** | **87.29** | **92.76** | **114** |

Table 5: Comparison with other interactive segmentation methods that integrate 3D information.

### A.5 ADDITIONAL VISUALIZATION OF ATTENTION WEIGHTS IN ORDER-AWARE ATTENTION

In this section, we provide more visualizations of the attention weights to show the importance of proposed order-aware attention, as shown in Figure 7 and discussed in 4.5.

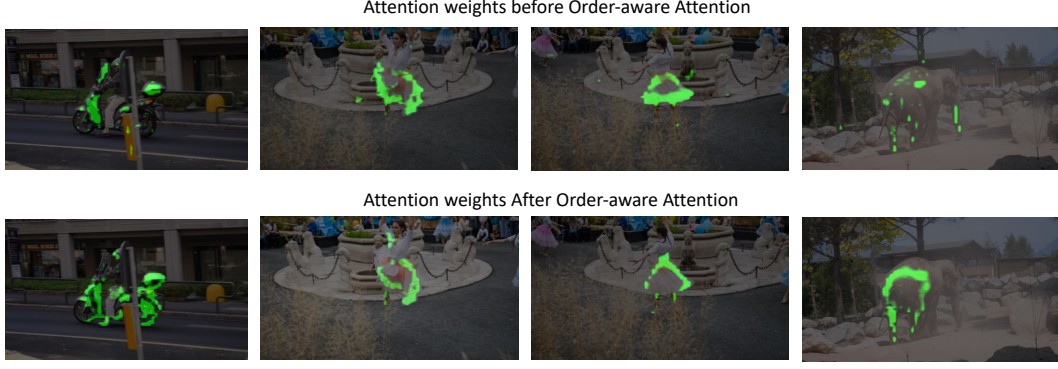

Figure 7: Visualization of attention weights before and after applying order-aware attention.

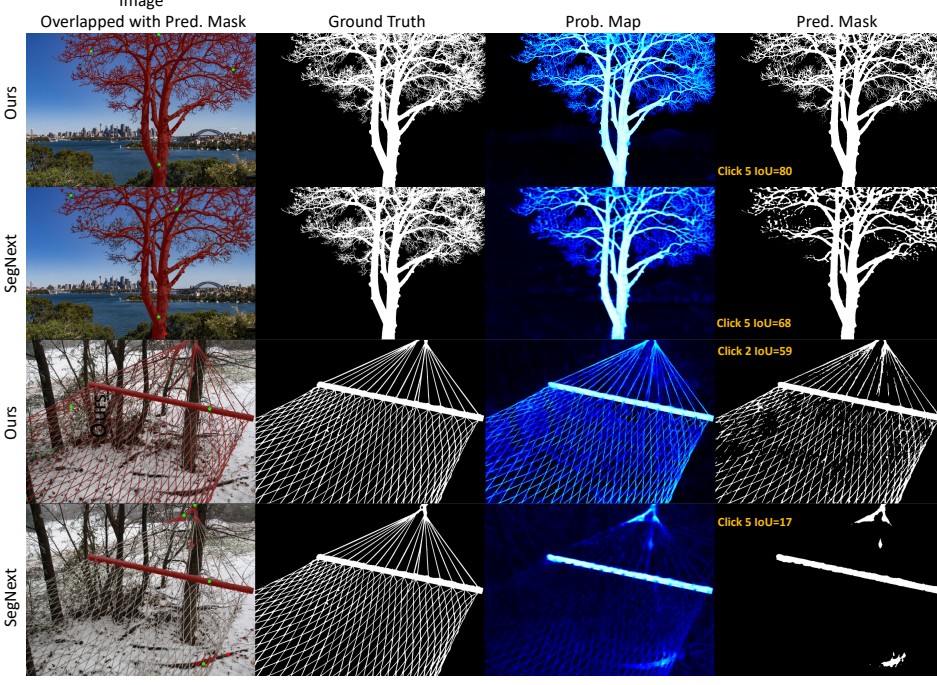

Figure 8: Qualitative results for more challenging cases with multiple clicks. Green dots mean the positive clicks, and red dots are the negative clicks.

### A.6 QUALITATIVE RESULTS FOR MORE CHALLENGING CASES

We provide more qualitative results of challenging cases from the HQSeg44K (Ke et al., 2024) dataset as displayed in Figure 8 and Figure 9.

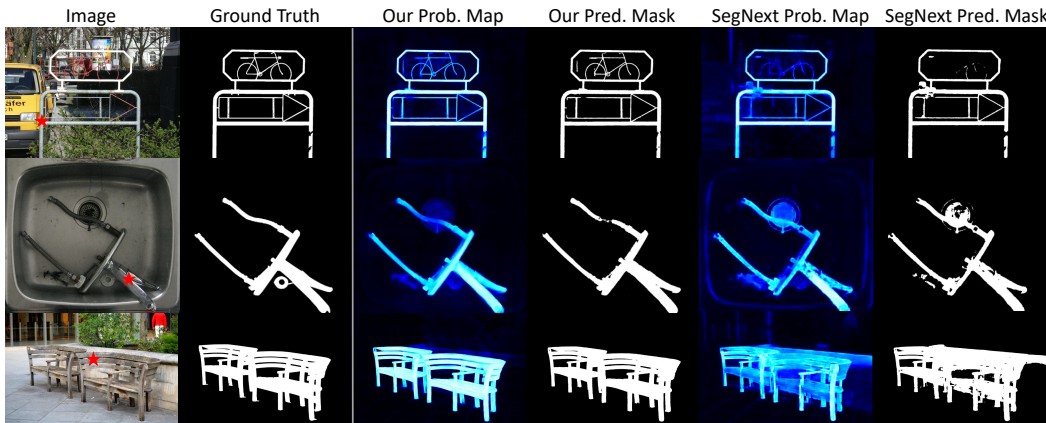

Figure 9: Qualitative results for more challenging cases with only one click. The red star represents the first click.

### A.7 ANALYSIS OF MULTI-ROUND INTERACTIONS

Interactive segmentation is a multi-round task. The goal is to minimize the user interactions (i.e., interaction round number) and achieve a good quality segmentation mask. Hence, we evaluate the multi-round interaction performance of our method. Figure 10 illustrates that while both our method and SegNext produce some false positives in the background after the first click, our method eliminates the entire background, including regions inside the bike and adjacent to the human head, with only a single negative click. In contrast, SegNext requires 10 clicks to remove these background false positives and still can not get satisfied segmentation for the entire target. We also provide more multi-round interaction results in Figure 12. For the challenging cases, the multi-round interaction results are displayed in Figure 11 and Figure 13.

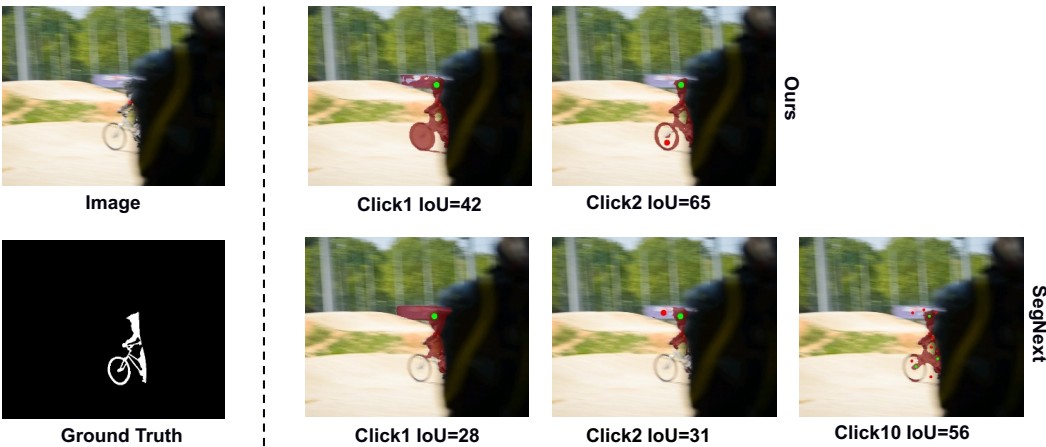

Figure 10: Multi-round interaction comparison. Green dots mean the positive clicks, and red dots are the negative clicks.

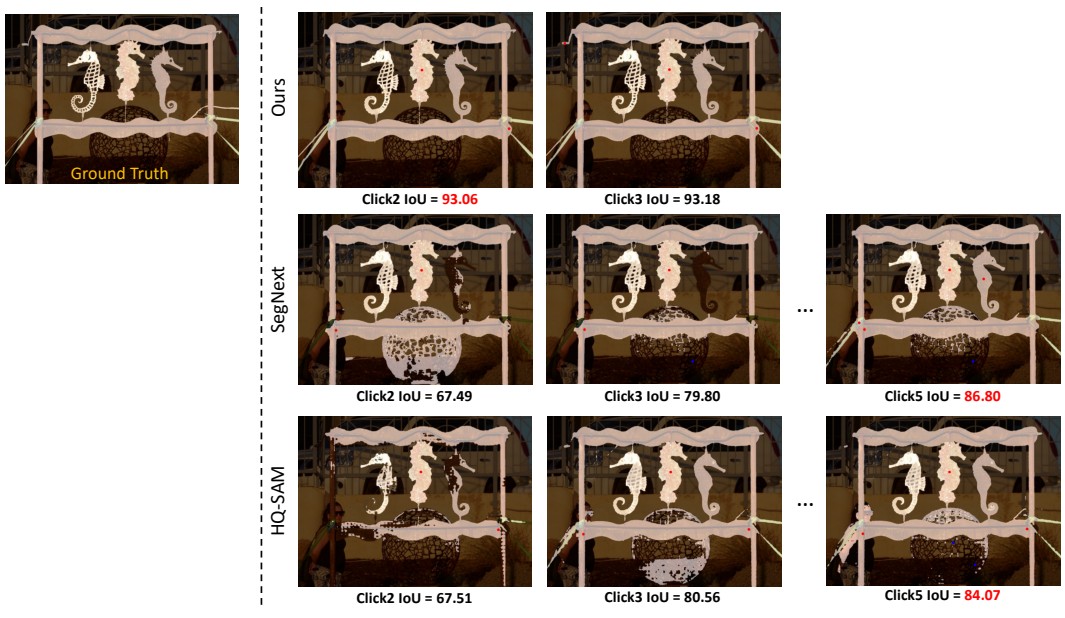

Figure 11: Multi-round interaction comparison for a challenging case.

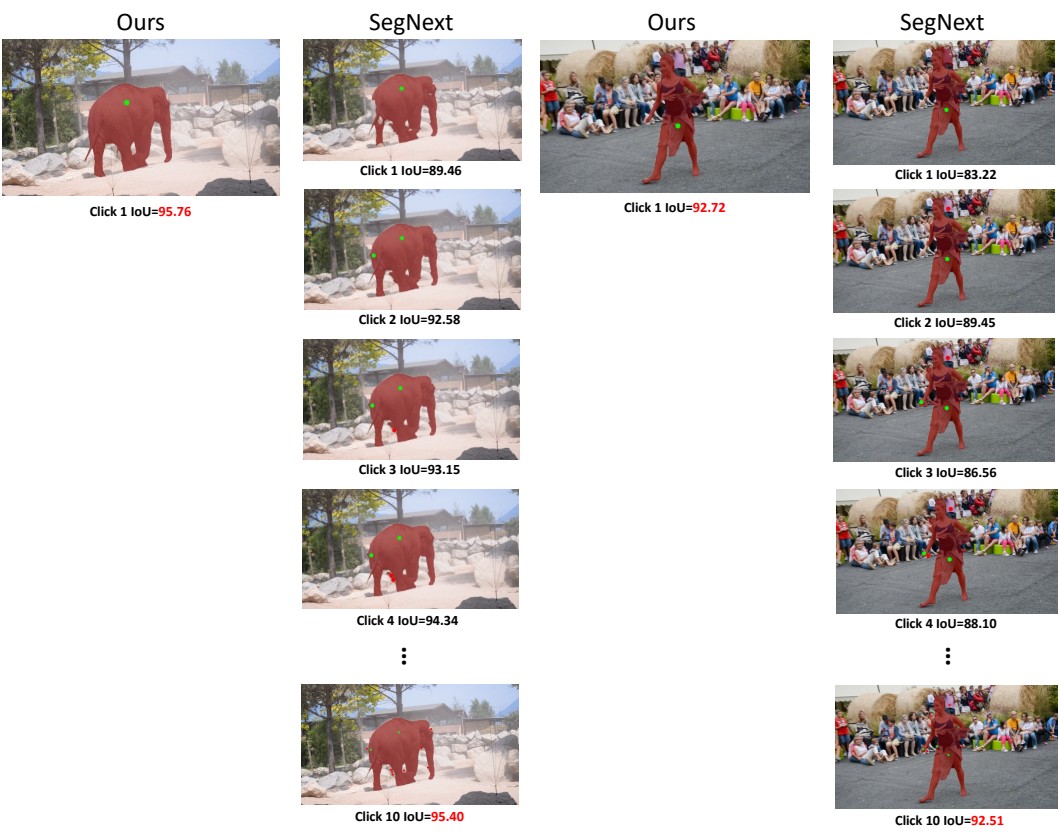

Figure 12: Multi-round interaction comparison. Green dots mean the positive clicks, and red dots are the negative clicks.

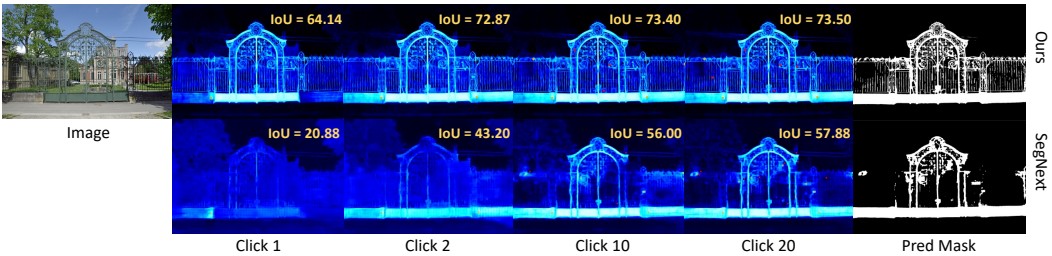

Figure 13: Multi-round interaction comparison for a challenging case.

## A.8 FAILURE CASES

In this section, we discuss some failure cases of our method. As shown in Figure 14, our method sometimes struggles to accurately predict the segmentation masks if thin objects occlude our target. This can be seen in the first example, where thin branches and leaves occlude the target bus and in the second and third examples, where the grass occludes the target human.

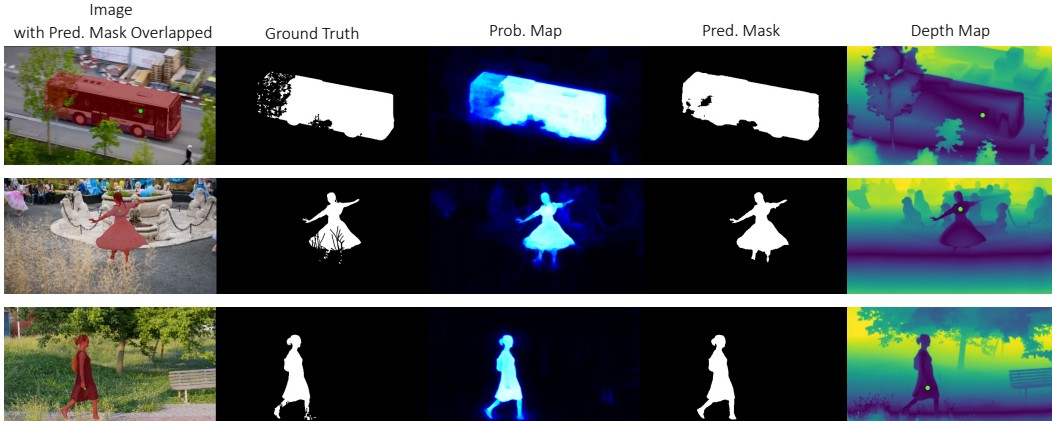

Figure 14: Failure Cases. Green dots are the first positive clicks.

## A.9 IMPACT OF DEPTH MAP ON MODEL PERFORMANCE

To evaluate the impact of the quality of the depth map used on our model's performance, we adopt three commonly used depth prediction models: DepthAnything V2 (Yang et al., 2024b), DepthAnything V1 (Yang et al., 2024a), and ZoeDepth (Bhat et al., 2023). As illustrated in Fig. 15, DepthAnything V2 produces the most fine-grained details, such as the girl's fingers and animal's tail, while the other two methods generate lower-quality depth maps with some possible depth prediction errors. Note that these three models generate depth maps in different quality levels, allowing us to get better insights on our model's robustness to the depth quality.

| Methods | Depth Model | NoC90 ↓ | 5-mIoU ↑ |
|---|---|---|---|
| SegNext (Liu et al., 2024a) | - | 4.43 | 91.87 |
| OIS | DepthAnythingV2 (Yang et al., 2024b) | 3.80 | **92.76** |
| OIS | DepthAnythingV1 (Yang et al., 2024a) | 3.78 | 92.69 |
| OIS | ZoeDepth (Bhat et al., 2023) | **3.75** | 92.75 |

Table 6: Performance comparison on DAVIS using depth maps from different depth prediction models.

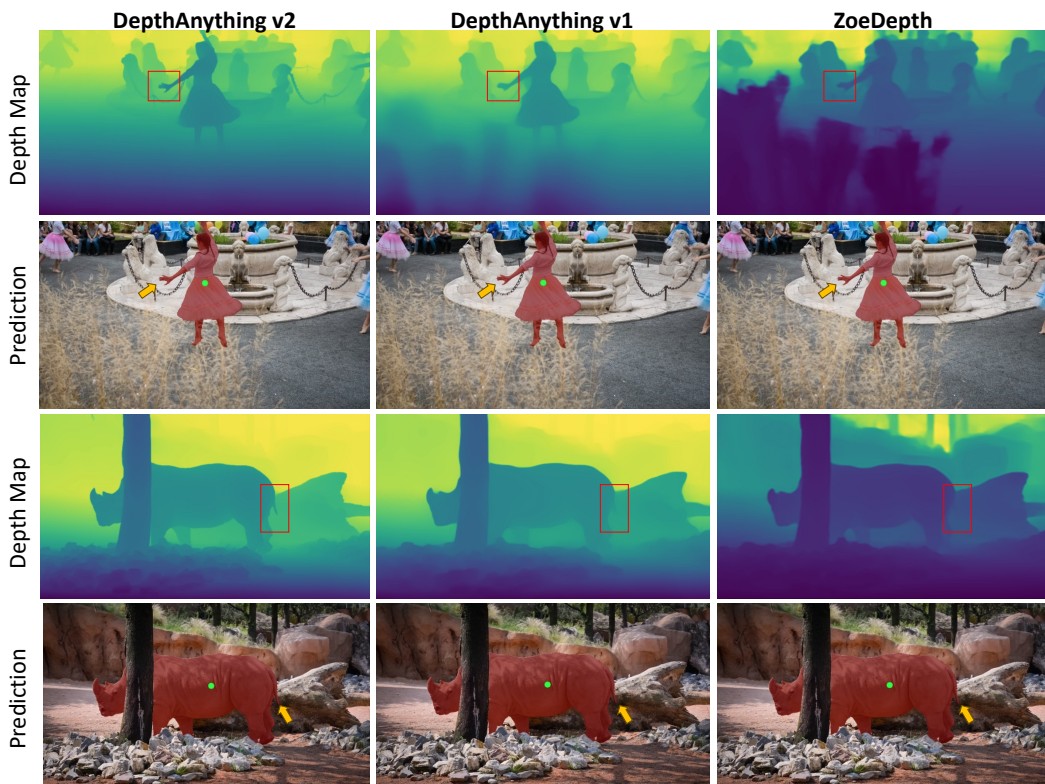

Figure 15: Qualitative results of our model with different depth map generation models. Green dots are the first positive clicks.

We compared our model's performance on the DAVIS (Perazzi et al., 2016) dataset using depth maps from these three models. The results, presented in Table 6, show minimal performance variation, with our method consistently outperforming the current SOTA method, SegNext (Liu et al., 2024a). This is because the depth map is used solely to generate the order map, which guides the model's understanding of relative depth between objects. Even a lower quality depth map (DepthAnything V1) has little impact on the final segmentation performance. More importantly, our proposed object-aware attention module is able to negate the effects of erroneous order maps caused due to erroneous depth maps.

Qualitative results are provided in Fig. 15. It is observed that even though the qualities of the depth map are different, the prediction masks are nearly the same, especially the regions that depth maps have significant different (girls' hand and animal's tail), which indicates the robustness of our model with different depth map source.

Moreover, it is important for the model to remain robustness when meeting the depth prediction errors. In Fig. 16, we put two examples which contain depth prediction error. In the first case, the dancer's hat is incorrectly blending into the background audience in the predicted depth map. However, our model correctly recovers the error and accurately segments the hat. Additionally, in the second case, the duck's boundary closely mixes with the neighboring water in terms of depth, while our model successfully separates the duck from the water in the segmentation prediction. This robustness is attributed to our proposed object-aware understanding module which ensures that our model comprehends the object as a whole, enabling it to handle depth prediction errors effectively.

Besides, it is noted that our model also achieve great performance in scenarios where significant depth variation exists within the target object. As demonstrated in Fig. 17, the target objects exhibit considerable depth variation, yet our model consistently delivers accurate and high-quality segmentation. This stems from the proposed object-aware understanding module plays a key role

in ensuring the model perceives the target object as a whole, enabling it to handle significant depth variation effectively.

In summary, the key role of depth in our approach is to address complex scenarios where distinguishing the target from the background is challenging, as demonstrated in Fig. 1 and Fig. 4. In these scenarios, our model significantly outperforms existing methods. For cases where depth is less critical (Fig. 17), we show that our model remains robust and unaffected by potential negative effects of depth. This demonstrates that our model can effectively solve challenging cases while maintaining strong performance in standard scenarios, highlighting its superior performance and robustness.

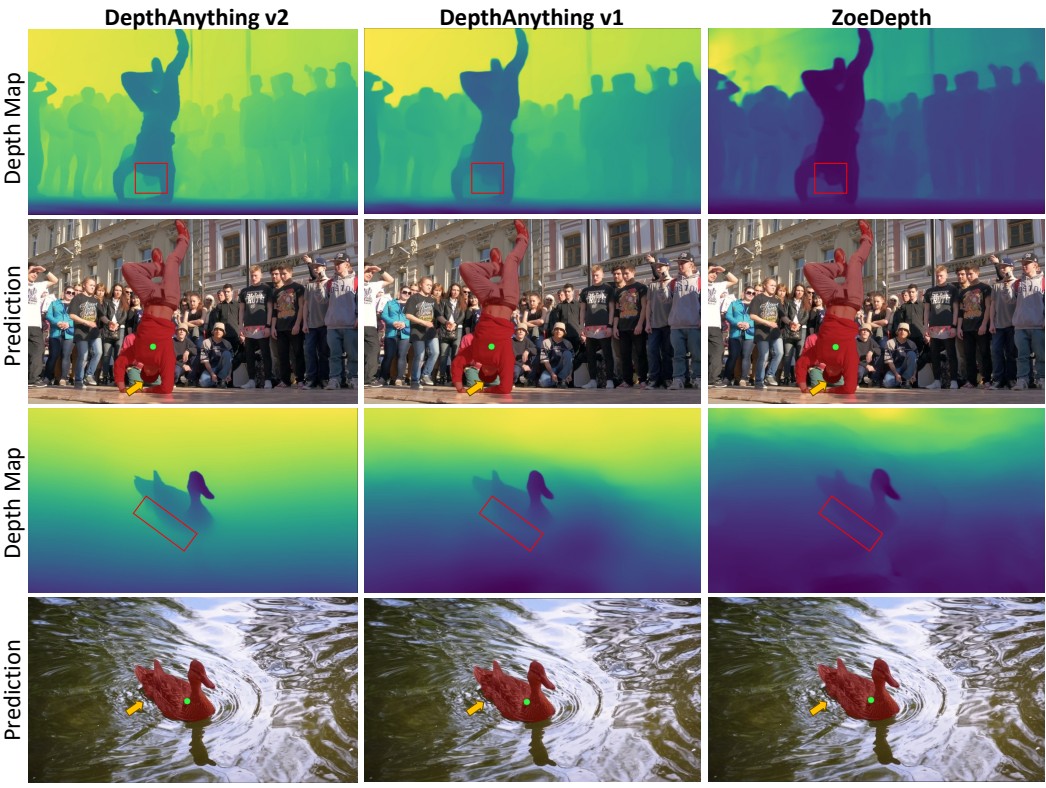

Figure 16: Qualitative results of our model in scenarios with depth prediction errors or targets with depth similar to neighboring objects. Green dots are the first positive clicks.

### A.10 ABLATION STUDY ABOUT THE ORDERING OF ORDER/OBJECT-AWARE UNDERSTANDING MODULES

Here, we discuss the effect of the sequence order of order-aware understanding and object-aware understanding module. We conduct the comparison experiment on DAVIS (Perazzi et al., 2016) dataset. The results in Table 7 show that having the order-aware understanding module first, followed by the object-aware understanding module slightly decreased performance than our current sequence (object-aware understanding module first, followed by the order-aware understanding module), indicating the effectiveness of our sequence choice.

### A.11 ABLATION STUDY ON HQSEG44K DATASET

To further prove the importance of each our proposed module, here we conduct the ablation experiment on HQSeg44K (Ke et al., 2024) dataset. The results, as displayed in Table 8, are consistent with the findings from the ablation study on the DAVIS dataset, as shown in Table 4, reaffirming that each proposed module plays a crucial role in enhancing overall performance.

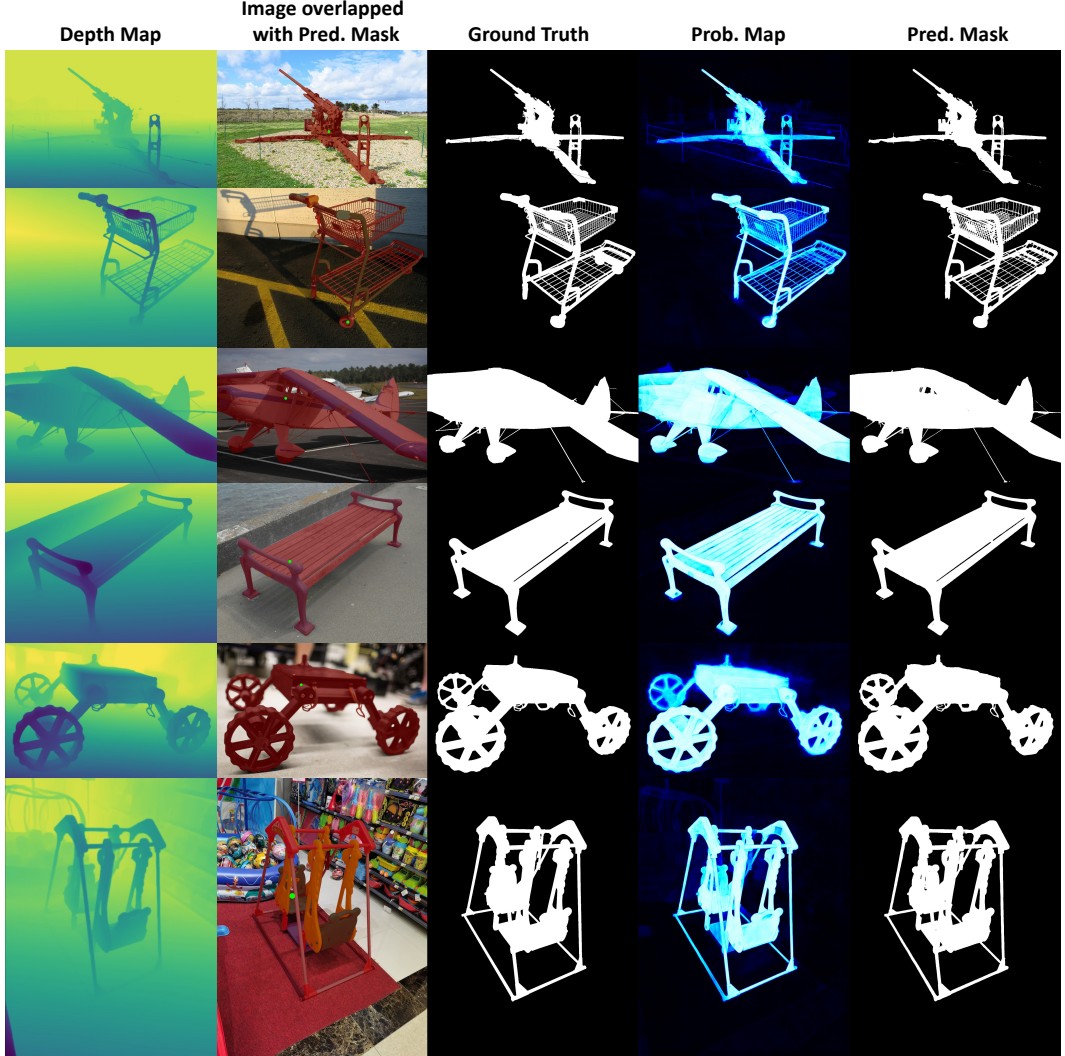

Figure 17: Qualitative results for cases with targets spanning a large depth range. Green dots are the first positive clicks.

| Methods | NoC90 ↓ | NoC95 ↓ | 1-mIoU ↑ | 5-mIoU ↑ |
|---|---|---|---|---|
| order-aware understanding first | 3.84 | 9.41 | 85.74 | 92.49 |
| object-aware understanding first (current) | **3.80** | **8.59** | **87.29** | **92.76** |

Table 7: Performance comparison on DAVIS with different sequence ordering of object and order-aware understanding modules.

| Methods | NoC90 ↓ | 5-mIoU ↑ |
|---|---|---|
| Full | 3.95 | 93.78 |
| w/o order | 4.87 (+0.98) | 92.49 (-1.29) |
| w/o object | 4.23 (+0.28) | 93.28 (-0.5) |
| w/o sparse | 5.23 (+1.28) | 90.80 (-2.98) |
| w/o dense | 4.97 (+1.02) | 91.75 (-2.03) |

Table 8: Ablation experiments on HQSeg44K.

## A.12 ABLATION STUDY FOR ORDER MAP WITH POSITIVE OR NEGATIVE CLICKS ALONE

We include an additional ablation study on DAVIS (Perazzi et al., 2016) dataset here to analyze the design of the order map. Table 9 shows that removing either the positive click order map or the negative click order map leads to a performance drop, confirming the effectiveness of combining both. Interestingly, we observe the following:

**Impact of Positive Click Order Map** When only the negative click order map is used, the 1-mIoU metric decreases more significantly. This suggests that the positive click order map is particularly beneficial during the first click, as the first click is always a positive click.

**Impact of Negative Click Order Map** When only the positive click order map is used, the NoC and 5-mIoU metrics see a larger decrease. This indicates that the negative click order map becomes more important as additional clicks are made. This is because subsequent clicks mainly involve adjustments and background removal, which rely heavily on negative clicks.

| Methods | NoC90 ↓ | NoC95 ↓ | 1-mIoU ↑ | 5-mIoU ↑ |
|---------|---------|---------|----------|----------|
| pos+neg | **3.80** | **8.59** | **87.29** | **92.76** |
| pos | 4.36 | 9.89 | 85.68 | 92.04 |
| neg | 4.01 | 9.24 | 84.17 | 92.37 |

Table 9: Ablation experiments for order with positive or negative clicks alone.

## A.13 ABLATION STUDY OF IMAGE ENCODER BACKBONE

To show the robustness of our model to different backbones, we conduct an ablation study on the HQSeg44K (Ke et al., 2024) dataset with different image encoder backbone. We replace the DepthAnything V2 backbone (Yang et al., 2024b) with an MAE pretrained ViT backbone (He et al., 2022) to be consistent with prior SOTA methods, SegNext (Liu et al., 2024a), InterFormer (Huang et al., 2023), and SimpleClick (Liu et al., 2023). Table 10 shows that our method still significantly outperforms the other methods with MAE ViT backbone. Furthermore, Table 11 highlights that the performance gains from our proposed order and object-aware attention far exceed those from switching backbones, representing the effectiveness and importance of our proposed methods.

| Methods | Backbone | NoC90 ↓ | NoC95 ↓ | 5-mIoU ↑ |
|---------|----------|---------|---------|----------|
| SimpleClick (Liu et al., 2023) | MAE ViT-B | 7.47 | 12.39 | 85.11 |
| InterFormer (Huang et al., 2023) | MAE ViT-B | 7.17 | 10.77 | 82.62 |
| SegNext (Liu et al., 2024a) | MAE ViT-B | 5.32 | 9.42 | 91.75 |
| OIS | MAE ViT-B | **4.41** | **8.01** | **93.12** |

Table 10: Comparison of performance using the same backbone with other SOTA methods.

| Methods | Backbone | NoC90 ↓ | NoC95 ↓ | 5-mIoU ↑ |
|---------|----------|---------|---------|----------|
| OIS w/o order+object | MAE ViT-B | 5.54 | 9.57 | 90.58 |
| OIS | MAE ViT-B | 4.41 | 8.01 | 93.12 |
| OIS w/o order+object | DepthAnythingV2 ViT-B | 5.23 | 8.91 | 90.80 |
| OIS | DepthAnythingV2 ViT-B | 3.95 | 7.50 | 93.78 |

Table 11: Comparison of performance improvement of order and object-aware attention with the same backbone.

A.14  PERFORMANCE COMPARISON TRAINED ON COCO

Since tradional interactive segmentation methods, including RITM (Sofiiuk et al., 2022), FocalClick (Chen et al., 2022), SimpleClick (Liu et al., 2023), and InterFormer (Huang et al., 2023) are purely trained on COCO+LVIS dataset (Lin et al., 2014; Gupta et al., 2019), to have a fair comparison with them, we also train our model using only COCO+LVIS dataset. In Table 12, we provide the comparison results on HQSeg44K (Ke et al., 2024) and DAVIS (Perazzi et al., 2016) dataset. The results demonstrate that our method still outperforms other methods by a large margin, which indicates the effectiveness of our model.

| | HQSeg44K | | | DAVIS | | |
| --- | --- | --- | --- | --- | --- | --- |
| | NoC90 ↓ | NoC95 ↓ | 1-mIoU ↑ | NoC90 ↓ | NoC95 ↓ | 1-mIoU ↑ |
| RITM (Sofiiuk et al., 2022) | 10.01 | 14.58 | 36.03 | 5.34 | 11.45 | 72.53 |
| FocalClick (Chen et al., 2022) | 7.03 | 10.74 | 61.92 | 5.17 | 11.42 | 76.28 |
| SimpleClick (Liu et al., 2023) | 7.47 | 12.39 | 65.54 | 5.06 | 10.37 | 72.90 |
| InterFormer (Huang et al., 2023) | 7.17 | 10.77 | 64.40 | 5.45 | 11.88 | 64.40 |
| OIS | **5.16** | **9.18** | **85.36** | **4.41** | **9.87** | **87.21** |

Table 12: Performance comparison with methods trained on COCO+LVIS.

