# OpenReview forum: "Order-aware Interactive Segmentation"
_ICLR.cc/2025/Conference — ICLR 2025 Poster_

### Official Review · Reviewer_DpQL · 2024-10-24

**Soundness:** 3
**Presentation:** 3
**Contribution:** 3
**Rating:** 8
**Confidence:** 4

**Summary:**

This work studies the interactive segmentation task. Based on the prior knowledge that ground and background objects are located at different depths, this work proposes a new framework named OIS. OIS effectively takes advantage of a corresponding depth map via the proposed order- and object-aware attention. Experiments demonstrate that OIS effectively improves segmentation performance by fewer clicks and boosts inference speed.

**Strengths:**

The manuscript is clearly written and easy to follow. The whole framework is built upon the fact that different objects should be located at various depths in the original 3D scenes, which is promising. Both qualitative and quantitative comparisons demonstrate its superior performance.

**Weaknesses:**

Overall, I feel quite OK with this work, and there are no MAJOR weaknesses. See below for several suggestions and typos.

- It is better to visualize the five clicks in Fig. 1 for an improved presentation.
- Paragraph 2, Sec. 1, Page 1: Redundant '(' before 'RITM'.
- Paragraph 2, Sec. 1, Page 2: Better to delete 'However' before 'current methods fail to ...'.
- The blue dots in Fig. 3 are inconspicuous. Please consider another conspicuous color.

**Questions:**

Please refer to the WEAKNESSES part.

---

> ### Author Response · Authors · 2024-11-21
>
> Dear Reviewer DpQL,
>
> Thank you for your encouragement and detailed suggestions. We have revised the paper accordingly. We are thrilled that you found our work “promising” and with “superior performance”!

---

### Official Review · Reviewer_pFaJ · 2024-11-01

**Soundness:** 3
**Presentation:** 3
**Contribution:** 3
**Rating:** 5
**Confidence:** 5

**Summary:**

Authors propose the order-aware interactive segmentation (OIS) to explicitly encode the relative depth between objects into order maps. Authors introduce a novel order-aware attention, where the order maps seamlessly guide the user interactions (in the form of clicks) to attend to the image features. Authors further present an object-aware attention module
to incorporate a strong object-level understanding to better differentiate objects with similar order. OIS achieves state-of-the-art performance, improving mIoU after one click by 7.61 on the HQSeg44K dataset and 1.32 on the DAVIS dataset

**Strengths:**

1.	OIS can distinguish target objects based on their relative depths from one another and better differentiate objects with similar order.
2.	OIS improves the computational efficiency.
3.	OIS achieves competitive performance

**Weaknesses:**

1. The Concept of Order. Authors should reevaluate the current definition of 'order,' which is often misleading. The term 'order' should pertain to the sequence of click instead of depth map in interactive segmentation.
2. Insufficient ablation study. More ablation studies are required to clarify the reason of performance improvement, for example the order of object-level attention and order-level attention, and the different pretrained weights of backbone, such as Depth Anything V1.
3. Unfair comparison. OIS adopts Depth Anything V2 as backbone, while Depth Anything V2 requires much more (about 6x) pretraining images than SAM and other backbone. And MM-SAM (Table 5 in Appendix) with Depth Anything V2 achieves similar performance with OIS w/o order and object attention (Table 4). The comparison is unfair. Authors should conduct more experiments to further verify the effectiveness of order map. For example, authors can freeze the backbone of SAM and train an additional depth head to obtain the depth map. Then authors train the object and order attention to verify that the improvement comes from the order map instead of better pretrained backbone.
3. Limited novelty. OIS integrates the object attention from Cutie and depth map into interactive segmentation. However, the novelty is somewhat constrained.

**Questions:**

1. Using the same order map for all positive points may be too naive and in may condition, the depth values of different positive points maybe different.

---

> ### Author Response · Authors · 2024-11-21
>
> Dear Reviewer pFaJ,
>
> Thank you for your thorough review and valuable suggestions. We have tried to address each of your concerns below.
>
> > 1. The concept of order
>
> In this paper, we have clearly defined “order” as “**the relative depth between objects in a scene**”. It does not relate to the sequence of clicks. However, we are happy to revise the paper if you have a suggestion for a different word that represents the relative depth between objects.
>
> > 2. (a) Ablation study: sequence of object and order-aware attention modules
>
> Our choice of this sequence is intuitive. The object-aware understanding module is placed before the order-aware understanding module to ensure that the model **first develops a clear notion of the target object and locates it accurately**. Following this, the order-aware understanding module is used to **refine the predictions by eliminating background regions and incorporating missed foreground regions with the help of the order maps**.
>
> We further study the significance of this sequence on the DAVIS dataset in the following Table A. The results show that having the order-aware understanding module first, followed by the object-aware understanding module slightly decreased performance than our current sequence (object-aware understanding module first, followed by the order-aware understanding module), indicating the **effectiveness of our sequence choice**.
>
> *Table A: Performance comparison on DAVIS dataset with different sequence ordering of object and order-aware understanding modules.*
>
> |  | NoC90 ↓ | NoC95 ↓ | 1-mIoU ↑ | 5-mIoU ↑ |
> | --- | --- | --- | --- | --- |
> | order-aware understanding first | 3.84 | 9.41 | 85.74 | 92.49 |
> | object-aware understanding first (current) | **3.80** | **8.59** | **87.29** | **92.76** |
>
> > 2. (b) Ablation study: different backbones
>
> To show the robustness of our model to different backbones, we conduct an ablation study on the HQSeg44K dataset, and tabulate the results in Tables B and C. We replaced the DepthAnything V2 backbone with an MAE pretrained ViT backbone to be consistent with prior SOTA, SegNext, InterFormer, and SimpleClick. Table B shows that **our method still significantly outperforms the other methods with MAE ViT backbone**. Table C highlights that **the performance gain from our proposed approach exceeds the performance gains from using a powerful backbone by a substantial margin.** This represents the effectiveness and importance of our proposed method.
>
> *Table B: Comparison of performance using the same backbone with other SOTA methods on HQSeg44K dataset.*
>
> |  | backbone | NoC90 ↓ | NoC95 ↓ | 5-mIoU ↑ |
> | --- | --- | --- | --- | --- |
> | SimpleClick | MAE ViT-B | 7.47 | 12.39 | 85.11 |
> | InterFormer | MAE ViT-B | 7.17 | 10.77 | 82.62 |
> | SegNext | MAE ViT-B | 5.32 | 9.42 | 91.75 |
> | OIS | MAE ViT-B | **4.41** | **8.01** | **93.12** |
>
> *Table C: Comparison of performance improvement of order and object-aware attention with the same backbone on HQSeg44K dataset.*
>
> |  | backbone | NoC90 ↓ | NoC95 ↓ | 5-mIoU ↑ |
> | --- | --- | --- | --- | --- |
> | OIS w/o order+object | MAE ViT-B | 5.54 | 9.57 | 90.58 |
> | OIS | MAE ViT-B | 4.41 | 8.01 | 93.12 |
> | OIS w/o order+object | DepthAnythingV2 ViT-B | 5.23 | 8.91 | 90.80 |
> | OIS | DepthAnythingV2 ViT-B | 3.95 | 7.50 | 93.78 |

---

> ### Author Response · Authors · 2024-11-21
>
> > 3. (a) Unfair comparison: Depth Anything V2 requires much more (about 6x) pretraining images than SAM and other backbone
>
> To address your concern about unfair comparison, we have provided 2 new tables (Tables B and C in the previous answer) that show our results using other backbones. The tables highlight that **our performance gains are mainly from our proposed order-aware and object-aware attention modules instead of the specific DepthAnything V2 backbone**.
>
> > 3. (b) Unfair comparison: “MM-SAM (Table 5 in Appendix) with Depth Anything V2 achieves similar performance with OIS w/o order and object attention (Table 4).”
>
> Please note that our main contribution lies in the object-aware and order-aware attention modules. We would respectfully like to point out that it is irrelevant to our work if OIS w/o order and object performs similar to MM-SAM. Our original OIS (that contains both the order and object-aware attention modules) obtains an NoC score of 3.80 and a 5-mIoU score of 92.76 on the DAVIS dataset as compared to an NoC score of 6.22 and a 5-mIoU score of 88.19 for MM-SAM with depth maps from DepthAnything V2. We think this is a significant improvement.
>
> > 3. (c) Freeze the backbone of SAM and train an additional depth head ... train the object and order attention to verify that the improvements don’t come from better backbones.
>
> Thanks for this insightful suggestion! We have adopted the MAE pretrained ViT-B backbone to be consistent with the current SOTA: SegNext, InterFormer, and SimpleClick. The results from Table C (in answer 2 (b)) indicate that **our performance improvements are mainly from the object and order-aware attention modules instead of a better backbone**.
>
> Our experiments with SAM backbone on our OIS model are in progress, and the results will be included in the revised version of our paper upon completion.
>
> > 4. Limited novelty
>
> Please see our response to all reviewers titled “Highlighting Our Technical Novelty”, especially points 1 and 2.
>
> > 5. Same order map for all positive points may be too naive... the depth values of different positive points may be different.
>
> Thanks for bringing up this important point. We have often encountered cases where the objects in a scene have variable depth values. While we completely agree with this comment, our powerful object-aware attention module is able to negate any discrepancy caused due to the simple order-map construction for positive clicks for objects with variable depth (Please see Fig. 17 and Section A.9 of our revised paper for more details). Preliminary experiments on using a more complex construction of order maps for positive clicks (one order map for every positive click) did not show a large improvement in the accuracy (please see the Table D below). Hence we chose to stick to a simpler design. However, please note that our model’s flexibility allows it to accommodate different order maps constructed for each positive click in the future for a more complex dataset if needed, just like our current design choice of using different order maps for different negative clicks.
>
> *Table D: Performance comparison of different order map designs for positive clicks.*
> |  | NoC90 ↓ | 1-mIoU ↑ |
> | --- | --- | --- |
> | unique order map for each positive click | **3.76** | 86.96 |
> | one order map for all positive clicks combined (current) | 3.80 | **87.29** |

---

> > ### Comment · Reviewer_pFaJ · 2024-11-25
> >
> > Thanks for your response, but some of my concerns still remain unsolved.
> >
> > (1) Firstly, how do you train your model with MAE? Or in another word, because the MAE backbone lacks the depth prediction ability, how do you implement your order map?
> >
> > (2) Secondly, the experiments with DepthAnything V1 backbone as mentioned in previous Weakness is still lacked.

---

> ### Author Response · Authors · 2024-11-24
> **A Gentle Reminder: Discussion Ends in Less Than 3 Days**
>
> Dear Reviewer pFaJ,
>
> This is a kind reminder that the discussion period will close on November 26, which is in **less than 3 days**. We hope our responses and clarifications have fully addressed your concerns. If you have any additional questions, we would be happy to provide further explanations. Thank you!

---

> ### Author Response · Authors · 2024-11-25
> **Further Clarification of our Additional Experiments**
>
> Dear Reviewer pFaJ,
>
> 1. We still use the Depth-AnythingV2 model for generating the depth maps to construct the order maps in this case; we have only replaced the backbone of our main architecture with MAE ViT-B backbone to be consistent with prior SOTA methods that use the MAE ViT-B backbone. Please note that the experiments in Table B and C are solely to show that **our method is robust to different backbones**, while keeping the depth prediction model consistent.
> 2. We are currently running an experiment that reuses the Depth-AnythingV1 encoder as our backbone. This is a time-intensive process, and we will include the results in the revised version of our paper. However, we hope that our experiments in Table B and C have adequately confirmed that our performance improvements are not due to a more advanced pretrained backbone.
>
>
>     Please also note that to show that **our method is robust to different depth-prediction models** while keeping the backbone consistent, we have conducted additional experiments with DepthAnything V1 [1] and ZoeDepth [2] as the depth prediction models as shown in Table E. In all the experiments in Table E, we have only changed the depth prediction model that generates the order maps, we have fixed the backbone of our model to our original setting. The results consistently show minor performance variations, with our method outperforming the current SOTA method, SegNext, in all cases. For further details, please refer to point 10 of our response to Reviewer rqn7 and point 1 of our response to Reviewer r4kn.
>
> Table E: Performance comparison on DAVIS using depth maps from different depth prediction models.
>
> |  | NoC90 ↓ | 5-mIoU ↑ |
> | --- | --- | --- |
> | SegNext | 4.43 | 91.87 |
> | OIS_Depthanythingv2 | 3.80 | **92.76** |
> | OIS_Depthanythingv1 | 3.78 | 92.69 |
> | OIS_ZoeDepth | **3.75** | 92.75 |
>
> Hopefully these explanations address your concerns. We are happy to answer any other questions that you have. Thank you very much for your time and effort!
>
> [1] Depth Anything: Unleashing the Power of Large-Scale Unlabeled Data, CVPR 2024
> [2] ZoeDepth: Zero-shot Transfer by Combining Relative and Metric Depth

---

> ### Author Response · Authors · 2024-11-30
>
> Dear Reviewer pFaJ,
>
> As suggested, we retrained our framework using the DepthAnything V1 pretrained backbone as the image encoder and the DepthAnything V1 decoder for depth prediction. It is important to note that DepthAnything V2 was fully replaced by DepthAnything V1 in both the image encoding and depth prediction processes. The results, shown in Table F, demonstrate that the **performance improvement achieved by our proposed approach significantly exceeds the gains obtained from utilizing a more powerful backbone**. This finding **aligns with the analysis presented in our response 2(b) and Table C**, which highlights the effectiveness and importance of our proposed method.
>
> *Table F: Comparison of performance improvement of order and object-aware attention with the different backbones on HQSeg44K dataset.*
>
> |  | backbone | NoC90 ↓ | NoC95 ↓ | 5-mIoU ↑ |
> | --- | --- | --- | --- | --- |
> | OIS w/o order+object | MAE ViT-B | 5.54 | 9.57 | 90.58 |
> | OIS | MAE ViT-B | 4.41 | 8.01 | 93.12 |
> | OIS w/o order+object | DepthAnythingV1 ViT-B | 5.46 | 9.80 | 90.91 |
> | OIS | DepthAnythingV1 ViT-B | 4.69 (-0.77) | 8.41 (-1.39) | 92.82 (+1.91) |
> | OIS w/o order+object | DepthAnythingV2 ViT-B | 5.23 | 8.91 | 90.80 |
> | OIS | DepthAnythingV2 ViT-B | 3.95 | 7.50 | 93.78 |

---

> ### Author Response · Authors · 2024-12-02
> **A Gentle Reminder: Discussion Ends in Less Than 1 Day**
>
> Dear Reviewer pFaJ,
>
> As the discussion period will end today, we sincerely hope our responses have answered your concerns. If there are any other questions, we are willing to make further clarification. We appreciate your engagement and constructive suggestions to our work. Thanks!

---

### Official Review · Reviewer_jpbL · 2024-11-02

**Soundness:** 3
**Presentation:** 3
**Contribution:** 3
**Rating:** 5
**Confidence:** 3

**Summary:**

This paper presents OIS, which explicitly encodes the relative depth between objects into order maps. It introduces an order-aware attention mechanism that guide the user interactions to attend to the image features by the order maps, and an object-aware attention module for better differentiation of objects with similar order. Experiments demonstrate the effectiveness and efficiency of OIS.

**Strengths:**

1. The OIS integrates relative depths between objects, into interactive segmentation, improving the model’s ability to distinguish target objects.
2. Experimental results show quite good effectiveness.

**Weaknesses:**

1. The difference between order-aware attention and object-aware attention proposed in this paper and masked attention seems to be only in the source of input mask, so both are slightly innovative.
2. In the comparative experiments of this paper, most of the traditional interactive segmentation methods (training on COCO) and SAM-based methods (training on SA-1B) have not been trained on HQSeg44K, while OIS is trained and tested on this dataset. The data quality of HQSeg44K is higher than that of COCO, so the fairness of the experiment is uncertain.

**Questions:**

The main motivation of this paper is to enhance the ability of interactive segmentation through depth information, but the paper seems to only show the cases where the depth of different positions of the same object is not very different, and there is little discussion about the effectiveness of the proposed method when the depth of different positions of the same object is very different (such as the bridge of DepthAnything visualization).

---

> ### Author Response · Authors · 2024-11-21
>
> Dear Reviewer jpbL,
>
> We appreciate your detailed reviews and valuable opinion. We have tried to address your concerns below.
>
> > 1. Difference between order-aware/object-aware attention and masked attention ... only in the source of the input mask, ... slightly innovative.
>
> Besides the source of the input mask, the way of mask guidance is also different. Please see our response to all reviewers titled “Highlighting Our Technical Novelty” (especially points 1(b) and 2(a)).
>
> > 2. Fair Comparison with methods trained on COCO
>
> The current state-of-the-art methods SegNext and HQ-SAM use HQSeg44K as their training dataset. So, to ensure fair comparison with these methods, we trained our method on the HQSeg44K dataset.
>
> For the fair comparison with other traditional methods solely trained on COCO, we train our model using COCO dataset only and compare with traditional interactive segmentation methods on HQSeg44K and DAVIS dataset. The results in Table A below show that **our model still outperforms other methods by a large margin**, which indicates the effectiveness of our model.
>
> *Table A: Performance comparison with methods trained on COCO.*
>
> |  | HQSeg44K |  |  | DAVIS |  |  |
> | --- | --- | --- | --- | --- | --- | --- |
> |  | NoC90 ↓ | NoC95 ↓ | 1-mIoU ↑ | NoC90 ↓ | NoC95 ↓ | 1-mIoU ↑ |
> | RITM | 10.01 | 14.58 | 36.03 | 5.34 | 11.45 | 72.53 |
> | FocalClick | 7.03 | 10.74 | 61.92 | 5.17 | 11.42 | 76.28 |
> | SimpleClick | 7.47 | 12.39 | 65.54 | 5.06 | 10.37 | 72.90 |
> | InterFormer | 7.17 | 10.77 | 64.40 | 5.45 | 11.88 | 64.40 |
> | Ours | **5.16** | **9.18** | **85.36** | **4.41** | **9.87** | **87.21** |
>
> > 3. Objects with variable depth
>
> Thanks for this valuable suggestion. We have **added qualitative results for objects with variable depth** in Section A.9 of our revised paper. As demonstrated in Fig. 17, **the target objects exhibit considerable depth variation, yet our model consistently delivers accurate and high-quality segmentations**. This robustness stems from the proposed object-aware understanding module that ensures the model perceives the target object as a whole, enabling it to handle significant depth variation effectively. In addition, please note that the “object-awareness” also allows our model to negate the effects of erroneous depth prediction (Please see section A.9 in the supplementary and our response to Reviewer r4kn - 2a and 2b).

---

> ### Author Response · Authors · 2024-11-24
> **A Gentle Reminder: Discussion Ends in Less Than 3 Days**
>
> Dear Reviewer jpbL,
>
> This is a kind reminder that the discussion period will close on November 26, which is in **less than 3 days**. We hope our responses and clarifications have fully addressed your concerns. If you have any additional questions, we would be happy to provide further explanations. Thank you!

---

> ### Author Response · Authors · 2024-12-02
> **A Gentle Reminder: Discussion Ends in Less Than 1 Day**
>
> Dear Reviewer jpbL,
>
> As the discussion period will end today, we sincerely hope our responses have answered your concerns. If there are any other questions, we are willing to make further clarification. We appreciate your engagement and constructive suggestions to our work. Thanks!

---

### Official Review · Reviewer_QSAV · 2024-11-02

**Soundness:** 4
**Presentation:** 3
**Contribution:** 3
**Rating:** 6
**Confidence:** 5

**Summary:**

1. This paper proposes order-aware attention, which integrates order (relative depths between objects) into interactive segmentation, improving the model’s ability to distinguish target objects based on their relative depths from one another.
2. This paper introduces object-aware attention to incorporate a strong understanding of objects.
3. This paper combines both dense and sparse integration of prompts, improving the alignment between the image and prompts while maintaining efficiency.
4. This work achieves good performance with lower latency.

**Strengths:**

1. The method looks very good for segmenting some difficult foreground objects (tennis rackets, bicycle wheels, etc.).
2. The order-aware attention module introduced in this paper is easy to accept and effective.
3. The framework of this paper is relatively concise and the implementation is easy to understand.

**Weaknesses:**

1. The method section of this paper is relatively easy to understand, but the motivation and problems to be solved are not concise enough. The author seems to want to solve many problems and propose many improvements, which can easily lead to readers not understanding why the author proposes certain techniques.
2. The experiments of this paper is insufficient, lacking results on typical interactive segmentation datasets, such as GrabCut, Berkeley, SBD, PascalVOC, etc. In addition, there is a lack of comparison with some recently published (CVPR 2024, etc.) interactive segmentation methods.
3. I admit the practical value of this paper, but I think it is more suitable to be published in CVPR rather than ICLR.

**Questions:**

1. Can the author connect and refine the motivation and the problem to be solved in this paper?
2. See Weaknesses section.

---

> ### Author Response · Authors · 2024-11-21
>
> Dear Reviewer QSAV,
>
> Thank you for the invaluable suggestions and for pointing out some critical questions. Below, we have tried to address each of your questions.
>
> > 1. (a) Relatively easy to understand method section
>
> Thank you very much for your positive comment!
>
> > 1. (b) The motivation and problems to be solved are not concise... want to solve many problems and propose many improvements.
>
> We have carefully summarized our motivation and proposed solution below.
>
> #### **Motivation:**
> We find that current interactive segmentation methods (SegNext, HQ-SAM, SAM, SimpleClick, etc) often fail to accurately separate target objects from the background in challenging cases with occlusions, multiple objects interacting with one another, and for thin and intricate objects with a vibrant background (Figs. 1, 4, 5, 8, 9). These issues occur due to a limited understanding of “order”, which we define to be the “relative depth of objects from one another in the scene”.
>
> #### **Solution:**
> To address the aforementioned issue, we propose the following solution:
>
> (a) We aim to incorporate the concept of “**order**” in our interactive segmentation model. This is performed by our order-aware attention module.
>
> (b) We incorporate the concept of “**objects**” to ensure that our model can distinguish objects belonging to the same “order”, and to ensure that our model is robust to erroneous order map construction (order maps depend on the quality of depth information available). The concept of “objects” is incorporated in our object-aware attention module.
>
> In addition, please see our response to all reviewers titled “Highlighting Our Technical Novelty”.
>
> > 1. (c) ... hard to understand why the author proposes certain techniques
>
> It would be very helpful if you could specify which technique needs to be better motivated. We will try our best to provide an intuition. In addition, please see our response to all reviewers titled “Highlighting Our Technical Novelty”.
>
> > 2. (a) Results on extra dataset
>
> We have used the same datasets for evaluation as the recent state-of-the-art interactive segmentation methods like SegNext, HR-SAM++, HR-SAM, and HQ-SAM. Moreover, compared to the four suggested datasets, our evaluation datasets are more challenging:  HQSeg44K contains a larger scale of data with much higher target annotation (44k images with over 1000 semantic classes); all images from the DAVIS dataset are real-world complex scenarios. Hence, these two datasets provide convincing evidence of our method’s efficacy.
>
> Additionally, following the suggestion, we evaluate our method on the GrabCut dataset, as shown in Table A. Note that none of the SAM-based methods have been evaluated on this dataset.  We aim to conduct the experiments on the other three suggested datasets and update the results in the revised version of the paper.
>
> *Table A: Performance comparison on GrabCut dataset.*
>
> |  | NoC85 ↓ | NoC90 ↓ | 1-mIoU ↑ |
> | --- | --- | --- | --- |
> | RITM | 1.46 | 1.56 | - |
> | FocalClick | 1.44 | 1.50 | - |
> | SimpleClick | 1.38 | 1.48 | - |
> | InterFormer | - | 1.36 | - |
> | MFP | 1.38 | 1.48 | - |
> | SegNext | 1.30 | **1.34** | 87.69 |
> | Ours | **1.28** | 1.44 | **89.62** |
>
> > 2. (b) “there is a lack of comparison with some recently published (CVPR 2024, etc.) ”
>
> Could you please point to these specific works?
>
> To the best of our knowledge, we have included the relevant state-of-the-art methods, including SegNext (CVPR 2024), HQ-SAM (NeurIPS 2023), SAM (ICCV 2023), InterFormer (ICCV 2023), and SimpleClick (ICCV 2023). However, we might have unintentionally missed some methods, owing to the overwhelmingly large number of papers in this domain. It would be great if you could please let us know what specific works to compare against.
>
> Based on your suggestion, we included discussion on some additional recent works (MFP [1] and GraCo [2] from CVPR 2024). We include MFP in Table A, however, MFP did not evaluate on HQSeg44k and their performance on the DAVIS dataset (NoC90: 5.32, NoC95: 11.27) is much lower than ours (NoC90: 3.80, NoC95: 8.59). GraCo, uses additional part-object training and operates under a different setting (multi-granularity), so we have excluded it from our comparison.
>
> [1] MFP: Making Full Use of Probability Maps for Interactive Image Segmentation, CVPR 2024
> [2] GraCo: Granularity-Controllable Interactive Segmentation, CVPR, 2024
>
> > 3. CVPR rather than ICLR
>
> Our paper fits in the “applications to computer vision, audio, language, and other modalities” subject area of ICLR 2025. To our knowledge,  many methods on interactive segmentation [3, 4], and depth-guided segmentation [5] have been previously published in ICLR, which closely relate to our work.
>
> [3] AGILE3D: Attention Guided Interactive Multi-object 3D Segmentation, ICLR 2024
> [4] Matcher: Segment Anything with One Shot Using All-Purpose Feature Matching, ICLR 2024
> [5] DFormer: Rethinking RGBD Representation Learning for Semantic Segmentation, ICLR 2024

---

> ### Comment · Reviewer_QSAV · 2024-11-23
>
> Thanks to the author's thoughtful response. This paper introduces depth information into the field of interactive segmentation. I recognize this contribution, so I will revise my score.

---

> ### Author Response · Authors · 2024-11-25
>
> Thank you very much for increasing the score! We are glad that we could address your concerns!

---

### Official Review · Reviewer_r4kn · 2024-11-02

**Soundness:** 2
**Presentation:** 3
**Contribution:** 2
**Rating:** 6
**Confidence:** 5

**Summary:**

The paper proposes order-aware interactive segmentation, which utilizes the extra relative depth information from pre-trained monocular depthanything v2 to generate the order maps. Then the order maps are used to guide the sparse embeddings to attend to the image features via mask attention. Object-aware attention is also used to help boost performance. Prompts are integrated via both the sparse and dense fusion. The paper validates its experiment design on HQSeg44K and DAVIS benchmark.

**Strengths:**

1. The paper has a good writing and structure, where the paper ideas and figures are easy to read and understand.

2. The paper validates its model design choice in Sec 4.5 and Table 4, which shows the performance gain brought by each proposed component clearly.

3. Using relative depth order to guide segmentation is an interesting idea, where the proposed order map considers both the positive and negative clicks.

**Weaknesses:**

1. The paper utilizes additional monocular depth prediction as input. In Table 3, does the computing cost of the depth model also get included? The paper should also study the impact of using various depth prediction models to the segmentation performance.

2. The robustness of the segmentation model to the errors brought by depth prediction network is not studied. When depth prediction makes large errors, how will it influence the segmentation model? Especially, when segmenting neighboring objects with close depth distance, how much can the depth model contribute?

3. Consider the importance of Table 4, besides DAVIS, an ablation experiment also on HQ-Seg44K should also be performed to have a more comprehensive understanding of each proposed component.

4. The paper has a limited tech novelty, where the order map guiding attention is borrowed from Mask2Former and the object-aware attention is from Cuite. The depth prediction is from a pretrained depthanything v2 model. Using depth to guide more accurate segmentation is also introduced in [a, b]. This makes the paper more like a combination of existing components and model designs.

[a] "Depth-Guided Semi-Supervised Instance Segmentation." arXiv preprint arXiv:2406.17413
[b] Unsupervised Semantic Segmentation Through Depth-Guided Feature Correlation And Sampling. CVPR, 2024.

5. The paper misses an ablation experiment on order maps designs considering both for positive and negative clicks. What if only considering the order maps for positive or negative clicks alone?

**Questions:**

What's the advantage of using depthanything v2's pretrained backbone in model's segmentation accuracy? besides saving parameters, how does it compare to the image encoder of SAM?

---

> ### Author Response · Authors · 2024-11-21
>
> Dear Reviewer r4kn,
>
> Thank you for your thorough feedback and constructive comments. We appreciate the opportunity to clarify the aspects you have mentioned. Our responses can be found below.
>
> > 1. (a) In Table 3, does the computing cost of the depth model also get included?
>
> Yes, we reported the SAT Latency metric [1] in Table 3, which includes the total time involved in prediction (encoding the image, **depth map generation**, encoding the clicks, and decoding the final segmentations). We have now further clarified this in Section 4.4 of our revised version. Please note that despite incorporating depth, our model achieves significantly lower latency while maintaining high segmentation accuracy, owing to the superior time-efficiency of recent SOTA depth prediction models like DepthAnything V2.
>
> [1] Rethinking Interactive Image Segmentation with Low Latency, High Quality, and Diverse Prompts, CVPR 2024
>
> > 1. (b) Impact of using various depth prediction models to the segmentation performance
>
> Table A below shows our results using different depth prediction models. Please note that our work **outperforms SegNext (the previous SOTA) by a substantial margin, with all depth prediction models we use**, highlighting the robustness of our carefully designed object-aware and order-aware attention modules. We also provide detailed discussion and visualizations (Fig.15, 16) in Section A.9 of the revised version of our paper.
>
> *Table A: Performance comparison on DAVIS using depth maps from different depth prediction models.*
>
> |  | NoC90 ↓ | 5-mIoU ↑ |
> | --- | --- | --- |
> | SegNext | 4.43 | 91.87 |
> | Ours (Depthanythingv2 [2]) | 3.80 | **92.76** |
> | Ours (Depthanythingv1 [3]) | 3.78 | 92.69 |
> | Ours (ZoeDepth [4]) | **3.75** | 92.75 |
>
> [2] Depth Anything V2, NeurIPS 2024
> [3] Depth Anything: Unleashing the Power of Large-Scale Unlabeled Data, CVPR 2024
> [4] ZoeDepth: Zero-shot Transfer by Combining Relative and Metric Depth
>
> > 2. (a) The robustness of the segmentation model to the errors brought by depth prediction
>
> In our previous answer, we have shown that **our model is robust to using different depth prediction networks**. This robustness is attributed to our **proposed object-aware understanding module which ensures that our model comprehends the object as a whole, enabling it to handle depth prediction errors effectively**. We present **additional qualitative results** in Section A.9 to highlight this. For instance, in Fig. 15, while ZoeDepth and DepthAnything V1 misinterpret the depth of the animal's tail, our model is robust to this error and can accurately segment the tail. In Fig. 16, despite a significant depth prediction error of all the depth models where the dancer’s hat blends into the background audience, our model correctly recovers the error and accurately segments the hat.
>
> Also, please note that the key role of depth in our approach is to address complex scenarios where distinguishing the target from the background is challenging, as demonstrated in Fig. 1 and Fig. 4. In these scenarios, our model significantly outperforms existing methods. For cases where depth is less critical (Fig. 17), we show that our model remains robust and unaffected by potential negative effects of depth. This demonstrates that **our model can effectively solve challenging cases while maintaining strong performance in standard scenarios**, which we have discussed more detailedly in Section A.9 of the revised paper.
>
> > 2. (b) “when segmenting neighboring objects with close depth distance, how much can the depth model contribute?”
>
> It is true that order maps may not be able to separate the foreground object from all background objects. However, we would like to point out that: 1) order maps can still help to eliminate distractions from most background objects. 2) our pipeline also leverages appearance / objectness features, on top of order maps. We show this qualitatively in Fig. 16, in which the depth prediction model fails to differentiate the duck’s boundary from the surrounding water body. However, our model’s “object-awareness” allows it to successfully segment the duck.
>
> > 3. Ablation on HQSeg44K dataset
>
> Thanks for the suggestion! Table B below shows the ablation results on the HQSeg44K dataset. These results are **consistent with the findings from the ablation study on the DAVIS dataset** (Table 4 of the main paper), reaffirming that each proposed module plays a crucial role in enhancing overall performance.
>
> *Table B: Ablation experiments on HQSeg44K.*
>
> |  | NoC90 ↓ | 5-mIoU ↑ |
> | --- | --- | --- |
> | Full | 3.95 | 93.78 |
> | w/o order | 4.87 (+0.98) | 92.49 (-1.29) |
> | w/o object | 4.23 (+0.28) | 93.28 (-0.5) |
> | w/o sparse | 5.23 (+1.28) | 90.80 (-2.98) |
> | w/o dense | 4.97 (+1.02) | 91.75 (-2.03) |

---

> > ### Author Response · Authors · 2024-11-21
> >
> > > 4. (a) Limited technical novelty
> >
> > Please see our response to all reviewers titled “Highlighting Our Technical Novelty”.
> >
> > > 4. (b) “order map guiding attention is borrowed from Mask2Former”
> >
> > We respectfully point out that our order-aware attention is different from the attention mechanism in Mask2Former. Please see 1(b) in our response to all reviewers titled “Highlighting Our Technical Novelty”.
> >
> > > 4. (c) “object-aware attention is from Cuite”
> >
> > We have clarified the difference between object-aware attention and Cutie in Section 3.3 of the main paper and 2(a) of our response to all reviewers titled “Highlighting Our Technical Novelty”.
> >
> > > 4. (d) “Using depth to guide more accurate segmentation is also introduced in [a, b].”
> >
> > Please see point 1 of our response to all reviewers titled “Highlighting Our Technical Novelty”.
> >
> > > 5. Ablation for order maps from positive or negative prompts alone
> >
> > Thanks for the suggestion! Table C below shows an additional ablation on the DAVIS dataset to analyze the design of the order map. We clearly see that **removing either the order map for positive clicks or the order maps for negative clicks will lead to a performance drop**, confirming the effectiveness of combining both. Please find more explanation and discussion in A.12 in supplementary materials.
> >
> > *Table C: Ablation experiments for order with positive or negative clicks alone.*
> >
> > |  | NoC90 ↓ | NoC95 ↓ | 1-mIoU ↑ | 5-mIoU ↑ |
> > | --- | --- | --- | --- | --- |
> > | pos+neg | **3.80** | **8.59** | **87.29** | **92.76** |
> > | pos | 4.36 | 9.89 | 85.68 | 92.04 |
> > | neg | 4.01 | 9.24 | 84.17 | 92.37 |
> >
> > > 6. Advantage of using DepthAnything V2's backbone & Comparison with other backbones
> >
> > DepthAnything V2 backbone can **capture more fine-grained details** when extracting the image features due to its pretraining on high-quality synthetic data. This is also beneficial for getting segmentation with enhanced details, for example, the tree sample of Fig. 8.
> >
> > To show the robustness of our model to different backbones, we conduct an ablation study on the HQSeg44K dataset, and tabulate the results in Tables D and E. We replace the DepthAnything V2 backbone with an MAE pretrained ViT backbone to be consistent with prior SOTA, SegNext, InterFormer, and SimpleClick. Table D shows that **our method still significantly outperforms the other methods with MAE ViT backbone**. Further, Table E highlights that the **performance gain from our proposed approach exceeds the performance gains from using a powerful backbone by a large margin**. This represents the effectiveness and importance of our proposed method.
> >
> > Please note that SAM is trained on a large set of real images where fine-grained structures are under-represented.  Hence, the SAM backbone is likely to underperform when segmenting thin and intricate structures. This is evident from Fig. 4 that shows the predictions from the SAM-based model, HQ-SAM. To further demonstrate this, experiments with SAM backbone on our OIS model are in progress, and the results will be included in the revised version of our paper upon completion.
> >
> > *Table D: Comparison of performance using the same backbone with other SOTA methods on HQSeg44K dataset.*
> >
> > |  | backbone | NoC90 ↓ | NoC95 ↓ | 5-mIoU ↑ |
> > | --- | --- | --- | --- | --- |
> > | SimpleClick | MAE ViT-B | 7.47 | 12.39 | 85.11 |
> > | InterFormer | MAE ViT-B | 7.17 | 10.77 | 82.62 |
> > | SegNext | MAE ViT-B | 5.32 | 9.42 | 91.75 |
> > | OIS | MAE ViT-B | **4.41** | **8.01** | **93.12** |
> >
> > *Table E: Comparison of performance improvement of order and object-aware attention with the same backbone on HQSeg44K dataset.*
> >
> > |  | backbone | NoC90 ↓ | NoC95 ↓ | 5-mIoU ↑ |
> > | --- | --- | --- | --- | --- |
> > | OIS w/o order+object | MAE ViT-B | 5.54 | 9.57 | 90.58 |
> > | OIS | MAE ViT-B | 4.41 | 8.01 | 93.12 |
> > | OIS w/o order+object | DepthAnythingV2 ViT-B | 5.23 | 8.91 | 90.80 |
> > | OIS | DepthAnythingV2 ViT-B | 3.95 | 7.50 | 93.78 |

---

> > > ### Comment · Reviewer_r4kn · 2024-11-25
> > >
> > > Thanks for the detailed clarification. Most of my concerns on the depth robustness have been addressed. Thus, I revised my score rating to 6 accordingly.

---

> ### Author Response · Authors · 2024-11-24
> **A Gentle Reminder: Discussion Ends in Less Than 3 Days**
>
> Dear Reviewer r4kn,
>
> This is a kind reminder that the discussion period will close on November 26, which is in **less than 3 days**. We hope our responses and clarifications have fully addressed your concerns. If you have any additional questions, we would be happy to provide further explanations. Thank you!

---

> ### Author Response · Authors · 2024-11-25
>
> Thank you very much for increasing the score! We are glad that we could address your concerns!

---

### Official Review · Reviewer_rqn7 · 2024-11-04

**Soundness:** 3
**Presentation:** 3
**Contribution:** 2
**Rating:** 6
**Confidence:** 4

**Summary:**

The paper presents Order-Aware Interactive Segmentation (OIS), combining order-aware and object-aware attention to improve segmentation accuracy and efficiency. Order maps help distinguish object depths, while foreground-background separation aids object differentiation. Using both dense and sparse prompt fusion, OIS achieves state-of-the-art results on HQSeg44K and DAVIS, boosting accuracy and speed.

**Strengths:**

1. The paper is well-written, with clear motivation.
2. It provides extensive experiments that convincingly demonstrate the proposed method's effectiveness.
3. The visualized analysis adds value by highlighting the necessity of incorporating depth information.
4. The method's performance surpasses current state-of-the-art methods.

**Weaknesses:**

1. The paper exceeds the ICLR 2025 10-page limit with 11 pages in the main text.
2. Overall, the technical contribution and novelty of this paper are incremental, as it mainly incorporates existing priors, such as depth maps and foreground-background masks, to enhance segmentation accuracy. Since these priors have already proven effective in general segmentation tasks, their success in interactive segmentation is unsurprising. I would encourage the authors to clarify the unique benefits these priors bring specifically to interactive segmentation.
3. The rationale for the ordering of Object-level Understanding before Order-level Understanding is unclear. Could the authors explain if this order enhances performance or if alternative orderings were tested?
4. The modules for Object-level and Order-level Understanding are connected sequentially. Why was a parallel structure not considered? Could it improve performance or efficiency?
5. Similar work (see [1]) also enhances foreground-background distinction in interactive segmentation; this should be discussed.
6. The paper lacks ablation studies on Mask Guidance within the object-aware and order-aware attention modules. For example, testing the effects of different Mask Guidance qualities would be beneficial.
7. It is unclear whether OIS can handle interactive segmentation for multiple objects simultaneously. Since Object-level Understanding relies on Mask Guidance from previous interactions, if the target object changes between clicks, OIS may struggle to shift focus to the new target due to constraints from the previous mask. This limitation could reduce the method's practical utility.

[1] Object Aware Contrastive Prior for Interactive Image Segmentation

**Questions:**

1. Would it be possible to show corresponding depth maps in Figure 6 to help readers better understand the importance of order-aware attention?
2. As the number of negative clicks increases, does the model generate a unique order map for each negative prompt? If so, this approach could result in significant memory usage. Has the author considered selectively merging these maps? Since repeated negative clicks are likely to target very localized areas, the corresponding depth maps would likely have high similarity, making unique order maps for each negative prompt potentially redundant.
3. Would lower-quality depth maps directly impact OIS performance, and conversely, would higher-quality maps improve it?

---

> ### Author Response · Authors · 2024-11-21
>
> Dear Reviewer rqn7,
>
> Thank you for your detailed review and insightful suggestions. Your feedback is invaluable for improving our work. Here, we address each of your concerns.
>
> > 1. Page limit
>
> We followed the official ICLR 2025 guidelines which state that “the optional ethics statement will not count toward the page limit”. Our main text follows the 10 page limit.
>
> > 2. (a) “Technical contribution and novelty of this paper are incremental”
>
> Please see our response to all reviewers titled “Highlighting Our Technical Novelty”.
>
> > 2. (b) Unique benefit of order in interactive segmentation
>
> For challenging cases in interactive segmentation, such as objects with occlusions, thin and intricate structures, etc, (refer to Fig. 1 and Fig. 4), users are often required to perform multiple interactions, i.e., redundant positive and negative clicks to capture the entire foreground and remove parts of the background. Our proposed design of using order maps to guide the attention to image features can help the model understand the relative depth between objects better. This enables the model to **distinguish the target from background effectively with significantly fewer user interactions** as shown in Table 1 and 2 of the original paper.
>
> > 3. Ordering of order-aware understanding and object-aware understanding
>
> Our choice of this sequence is intuitive. The object-aware understanding module is placed before the order-aware understanding module to ensure that the model **first develops a clear notion of the target object and locates it accurately**. Following this, the order-aware understanding module is used to **refine the predictions by eliminating background regions and incorporating missed foreground regions with the help of the order maps**.
>
> We further study the significance of this sequence on the DAVIS dataset in the following Table A. The results show that having the order-aware understanding module first, followed by the object-aware understanding module slightly decreased performance than our current sequence (object-aware understanding module first, followed by the order-aware understanding module), indicating the **effectiveness of our current sequence choice**.
>
> *Table A: Performance comparison on DAVIS dataset with different sequence ordering of object and order-aware understanding modules.*
>
> |  | NoC90 ↓ | NoC95 ↓ | 1-mIoU ↑ | 5-mIoU ↑ |
> | --- | --- | --- | --- | --- |
> | order-aware understanding first | 3.84 | 9.41 | 85.74 | 92.49 |
> | object-aware understanding first (current) | **3.80** | **8.59** | **87.29** | **92.76** |
>
> > 4. Parallel structure for order-aware and object-aware understanding modules
>
> We thank the reviewer for the suggestion. Incorporating this could be interesting, which would require significant effort on retraining and experiments. We plan to leave this for future work.
>
> However please note that our current choice of placing these modules is straightforward and intuitive (please see the previous answer for the reasoning behind this) which helps us achieve SOTA performance.
>
> > 5. Prior work also enhances foreground-background distinction
>
> Thanks for the suggestion. We have now discussed this work in the detailed related work section A.3 in our revised version.
>
> Please note that the suggested paper discovers foreground objects by training an additional network to predict foreground object features, and uses these object features as a prior input to the segmentation network. In contrast, we discover and refine foreground objects through our novel object-aware and order-aware attention modules, which is very different. On the DAVIS dataset, our method achieves an **88.05 1-mIoU**, outperforming the suggested paper's **80.02**, which indicates a **stronger foreground-background distinction capability in our approach**.

---

> > ### Author Response · Authors · 2024-11-21
> >
> > > 6. Ablation on mask guidance within the object-aware and order-aware attention modules
> >
> > Please note that it is non-trivial to introduce different mask guidance methods **within the object-aware and order-aware attention modules**. We are only aware of masked-attention [1] and its variant (foreground-background masked attention in Cutie [2]) to introduce mask-guidance in attention modules. We utilize the mask guidance from Cutie in our object-aware module. We propose a **novel order-aware cross attention** mechanism to use order maps for “soft” mask-guidance (assign higher attention to regions near the user-selected object and lower attention to farther regions) in the order-aware attention modules (Please see Section 3.2 for more details). We are not aware of other methods that integrate mask-guidance into attention modules.
> >
> > However, as suggested, we study the effects of different mask guidance methods in our overall pipeline (outside the attention modules) on the DAVIS dataset. Specifically, we trained two new encoders: one to encode the order maps (to replace order-aware attention) and the other to encode the previous segmentation masks (to replace object-aware attention). The encoded order maps and segmentation masks were concatenated with the image features and fused with the encoded prompts using traditional cross-attention. We call this way of mask-guidance “mask concat” in Table B below. The row “mask attention” refers to our original setting where we use the order-aware and object-aware attention mechanisms. The results demonstrate that **our design choice is significantly more effective as compared with naive mask-guidance.**
> >
> > *Table B: Performance comparison of different mask guidance methods.*
> >
> > |  | NoC90 ↓ | 1-mIoU ↑ | 5-mIoU ↑ |
> > | --- | --- | --- | --- |
> > | mask concat | 4.36 | 86.20 | 91.65 |
> > | mask attention | **3.80** | **87.29** | **92.76** |
> >
> > [1] Masked-attention mask transformer for universal image segmentation, CVPR 2022
> > [2] Putting the object back into video object segmentation, CVPR 2024
> >
> > > 7. Unclear if OIS can handle multiple objects, it lacks practical utility.
> >
> > We follow the same standard setting of interactive segmentation [3], like prior works (SAM, HQ-SAM, SegNext, SimpleClick, InterFormer, etc) which are all designed for single object segmentation. This type of interactive segmentation has been deployed in many practical applications successfully including medical annotation [4], video object tracking [5], etc. Hence, we disagree that this method lacks practical utility.
> >
> > Please note that our method can seamlessly be extended to segment multiple objects by sequentially segmenting each object like [6]. The image features are re-used, thereby incurring very little additional computational costs with more objects segmented.
> >
> > [3] RITM: Reviving Iterative Training with Mask Guidance for Interactive Segmentation, ICIP 2022
> > [4] Segment anything in medical images, Nature Communication 2024
> > [5] https://max810.github.io/xmem2-project-page/
> > [6] https://segment-anything.com/
> >
> > > 8. Add depth maps in Fig. 6
> >
> > Thanks for the suggestion. We have added the depth maps in Fig. 6 in the revised version of our paper.

---

> > > ### Author Response · Authors · 2024-11-21
> > >
> > > > 9. (a) Does the model generate a unique order map for each negative prompt?
> > >
> > > Yes. This is because the order maps are relative to the object on which the prompt/click is located, and each negative prompt can correspond to different objects in the background with different depth values.
> > >
> > > > 9. (b) Significant memory usage to construct an order map for every negative click
> > >
> > > In our experiments, the order maps are computed at a **low resolution (64×64) with very minor extra memory cost**. Most predictions can reach satisfactory results with under 10 clicks, making it unlikely for users to add any significant amount to memory usage due to excess number of clicks. Hence, the total computational cost typically remains reasonably low. For example, by using our method, the average number of user clicks per object in the DAVIS dataset to achieve high-quality segmentation (95% mIoU) is 8.59. An image that may require >20 clicks would already be impractical to interactively segment.
> > >
> > > > 9. (c) Consider selectively merging the order-maps for negative clicks
> > >
> > > Thanks for the suggestion. Please note that individual negative clicks can often point to different objects in the background, each having different depths. So, it is important to construct unique order maps to represent each negative click to better differentiate them. Given the low resolution of the order maps (64x64) discussed in the previous answer, this design is efficient and does not compromise our model’s ability to distinguish background objects at different depth levels. However, we do agree that based on the depth differences of the clicks, the pipeline can potentially be further optimized if we can selectively and adaptively combine them based on their differences in the depth values.
> > >
> > > > 9. (d) Repeated negative clicks are likely to target localized areas ... unique order maps potentially redundant.
> > >
> > > We agree that there can be potential redundancy in the order maps and this idea can be further explored. However, we want to point out that redundant order-maps incur only very minor additional computational cost without losing any potentially useful information (e.g. even when the depth difference between two clicks is small, it could reflect the user’s intention to correct a subtle segmentation error, in which case separate order maps can provide precise guidance to the segmenter). In addition, our model significantly reduces the number of negative clicks needed to eliminate false positives by effectively differentiating background objects. This point is elaborated in Section A.7 of the supplementary in the original paper, where we also provide qualitative examples to highlight this.
> > >
> > > > 10. Would lower-quality depth maps directly impact OIS performance, and conversely, would higher-quality maps improve it?
> > >
> > > We observed that **our model is robust to the quality of depth maps.**
> > >
> > > We conducted an ablation study using three commonly-used depth prediction models: DepthAnything V2 [7], DepthAnything V1 [8], and ZoeDepth [9]. DepthAnything V2 generates high quality predictions and preserves fine-grained details, however, we observe lower quality depth predictions from DepthAnything V1 and ZoeDepth as shown in Fig. 15 of our updated paper. We compared our model's performance on the DAVIS dataset using depth maps from these three models. The results, presented in the following Table C and Fig. 15, **show minimal performance variation, with our method consistently outperforming the current SOTA method**, SegNext. We believe this is because our object-aware attention module is able to negate the effects of erroneous order maps caused due to erroneous depth maps. We have discussed this point detailedly in Section A.9 of the revised version paper.
> > >
> > >
> > > *Table C: Performance comparison on DAVIS using depth maps from different depth prediction models.*
> > > |  | NoC90 ↓ | 5-mIoU ↑ |
> > > | --- | --- | --- |
> > > | SegNext | 4.43 | 91.87 |
> > > | OIS_Depthanythingv2 | 3.80 | **92.76** |
> > > | OIS_Depthanythingv1 | 3.78 | 92.69 |
> > > | OIS_ZoeDepth | **3.75** | 92.75 |
> > >
> > > [7] Depth Anything V2, NeurIPS 2024
> > > [8] Depth Anything: Unleashing the Power of Large-Scale Unlabeled Data, CVPR 2024
> > > [9] ZoeDepth: Zero-shot Transfer by Combining Relative and Metric Depth

---

> ### Author Response · Authors · 2024-11-24
> **A Gentle Reminder: Discussion Ends in Less Than 3 Days**
>
> Dear Reviewer rqn7,
>
> This is a kind reminder that the discussion period will close on November 26, which is in **less than 3 days**. We hope our responses and clarifications have fully addressed your concerns. If you have any additional questions, we would be happy to provide further explanations. Thank you!

---

> > ### Comment · Reviewer_rqn7 · 2024-11-26
> >
> > The authors' reply solved my doubts and I am willing to raise my rating to 6

---

> > > ### Author Response · Authors · 2024-11-26
> > >
> > > Thank you very much for increasing the score! We are glad that we could address your concerns!

---

### Author Response · Authors · 2024-11-21
**Highlighting Our Technical Novelty**

We thank all the reviewers for their thorough analysis and detailed feedback on our work.

  We are thrilled that they found our work **“interesting idea”** (Reviewer r4kn), **“good for segmenting some difficult foreground objects”** (Reviewer QSAV), having **“no major weaknesses”** (Reviewer DpQL), **“improve computational efficiency”** (Reviewer pFaJ), **“well-written, with clear motivation”** (Reviewer rqn7), having **“quite good effectiveness”** (Reviewer jpbL) and with **“superior performance”** (Reviewer DpQL).

Some reviewers have expressed their concerns regarding our technical novelty, which we carefully address below.

---

### Technical contributions of Order-Aware Interactive Segmentation (OIS):

1. We introduce a novel concept called “order”, in the context of interactive segmentation, which is defined as the **relative depth between objects** in a scene. This concept is important and **more meaningful than interpreting the absolute depth values** for the interactive segmentation task, because, intuitively, our model only needs to leverage the relative **ordering of objects**, i.e., whether some objects are closer or further to a reference object than others. This prevents our model from suffering due to the noise and scale differences of absolute depth values.


   > a. The concept of order allows us to seamlessly integrate positive and negative clicks into the interactive segmentation task through the construction of order maps. **This is different from naively integrating depth**, which results in suboptimal performance, as demonstrated by the MM-SAM results in Table 5 and Fig. 5 of the main paper.



   > b. We design a novel order-aware attention module that utilizes order maps to selectively attend to the image features. Unlike previous approaches [1,2,3], which have only used segmentation masks for masked attention, our method, to the best of our knowledge, is the **first to leverage other modalities (order maps) to guide attention** effectively. Note that, different from Mask2Former [1] and Cutie [2], our order-aware attention mechanism incorporates a “soft” mask guidance i.e., a continuous variant of masked cross attention. It prioritizes focus on closer objects (lower order) while gradually reducing focus on the further ones (higher order). Mask2Former [1] and Cutie [2], on the other hand, employ binary masks to either completely restrict attention to certain image regions, or to fully attend to the unblocked image regions.

---

2. We introduce the concept of “object” to ensure that our model can **distinguish objects belonging to the same order**, and to ensure that **our model is robust to erroneous order map construction** (order maps depend on the quality of depth information available).


   >a. The concept of “objects” is incorporated in our object-aware attention module.  This module is indeed similar to Cutie. However, unlike Cutie, we adapt this design for the interactive segmentation task, which is not a straightforward adaptation. Unlike Cutie, which initializes object embeddings randomly, **our method encodes foreground and background clicks as the object embeddings**, seamlessly allowing us to use the object-aware attention module to the interactive segmentation task. The encoded foreground and background clicks as object embeddings introduce a **discriminative notion of the target object** and enable the network to distinguish different objects with similar depth.

---

3. To seamlessly integrate both object and order-aware attention modules together meaningfully to yield the best results, we follow a sequential design of cascading the object-aware and order-aware attention modules one after the other. This design is intuitive and ensures that the model **first develops a clear notion of the target object** and locates it accurately. The order-aware attention module is then used to **refine the predictions with the help of the order maps**. This step is crucial to address challenging cases, i.e., objects with occlusions, thin and intricate structures, etc, as shown in Fig. 4-6, and Fig. 8-13.

---

4. To further improve our accuracy and speed, we combine two different approaches to integrate prompts: dense and sparse fusion. Prior interactive segmentation methods typically either use dense fusion (which is slow) or use sparse fusion (which is less accurate) and hence suffer from their respective limitations. In contrast, our approach ensures precise alignment between the image and the prompt for **high accuracy** (due to dense fusion) while maintaining **fast computational efficiency** (due to sparse fusion).

---

We have addressed the other concerns expressed by the reviewers one by one in separate comments. The reviewers' feedback has undoubtedly strengthened our paper, and we hope that our efforts have effectively addressed their concerns.

---

> ### Author Response · Authors · 2024-11-21
>
> ### References:
> [1] Masked-attention mask transformer for universal image segmentation, CVPR 2022
> [2] Putting the object back into video object segmentation, CVPR 2024
> [3] HODOR: High-level Object Descriptors for Object Re-segmentation in Video Learned from Static Images, CVPR 2022

---

### Author Response · Authors · 2024-12-03
**Revised Manuscript Updates and Additional Analysis for Reviewer Comments**

Dear Reviewers and Area Chairs,

We sincerely appreciate your insightful comments and valuable suggestions for our paper.  Your expert opinions have been invaluable towards the improvements we have made to our paper.

According to the points raised during the discussion, we have made the following updates in our revised paper:

1. Ablation study of image encoder backbone
    - We replaced our image encoder backbone with an MAE-pretrained ViT backbone, which is consistent with prior SOTA methods. Results in the **new Section A.13** confirm that our method significantly outperforms other methods with the same backbone, demonstrating that our performance improvements are primarily due to our proposed approach instead of a better backbone. This addresses concerns from Reviewers **r4kn** and **pFaJ**.

2. Impact of Depth Map on Model Performance
    - To address concerns raised by Reviewers **rqn7**, **r4kn**, and **jpbL**, we conducted additional ablation studies using various depth prediction models. These studies demonstrate the robustness of our model to variations in the depth map quality.
    - We provide additional qualitative results and discussion on this in **new** **Section A.9**.

3. Ablation study on the sequence of order/object-aware understanding modules
    - We have conducted experiments to validate our current sequence of the order-aware understanding module and the object-aware understanding module. We discuss the results in **Section A.10**, addressing the questions of Reviewers **rqn7** and **pFaJ**.

4. Fair Comparison with methods trained on COCO
    - To ensure fair comparisons, we trained our model solely on the COCO dataset and compared it with methods trained only on COCO. The results in the **new Section A.14** show that our model maintains a significant performance margin over other approaches, addressing the concerns raised by Reviewer **jpbL**.

5. Ablation study for order map with positive or negative clicks alone
    - We analyzed the impact of using order maps exclusively for positive or negative clicks. The results show that removing either leads to a performance drop. This is detailed in the **new Section A.12**, addressing questions from Reviewer **r4kn**.

6. Ablation Study on HQSeg44K dataset
    - To address the concern from Reviewer **r4kn**, we added an ablation study on the HQSeg44K dataset in **new Section A.11**. The results align with ablation study on the DAVIS dataset (in our original Table 4), reconfirming the effectiveness of each proposed module in our model.

7. Figures Revisions
    - We have updated **Fig. 1** to include the user prompts (positive and negative clicks) as recommended by Reviewer **DpQL**.
    - We have included depth maps in **Fig. 6,** as suggested by Reviewer **rqn7,** for better visualization.

8. Related Work Section
    - We have added and discussed the additional reference suggested by Reviewer **rqn7** in the **Detailed Related Work Section A.3.**

---

### Meta-Review · Area_Chair_RkW9 · 2024-12-16

**Metareview:**

The paper receives 4 positive and 2 negative ratings after rebuttal, with 3 upgraded scores. Initially, the reviewers had several concerns about technical motivation/contribution, ablation study, handling multiple objects, robustness to depth maps, and experimental fairness. In the post-rebuttal discussion period, three reviewers were satisfactory with the authors' comments and raised the rating. After taking a close look at the paper, rebuttal, and discussions, the AC agrees with reviewers' feedback of the proposed method being novel and effective to perform interactive segmentation. Therefore, the AC recommends the acceptance rating.

**Additional Comments On Reviewer Discussion:**

In the rebuttal, most critical concerns from the reviewer rqn7, r4kn, and QSAV, about technical contributions and more results (e.g., ablation study, robustness to depth maps) are well received by the reviewers. Moreover, for the reviewer jpbL and pFaJ who are mainly concerned about some experimental settings for fair comparisons, the discussions were not participated actively. Therefore, the AC took a close look at the rebuttal, discussions, and responses, in which the AC finds that the raised issues are addressed well by the authors.

---

### Decision · Program_Chairs · 2025-01-22

Accept (Poster)